# Bandit Task Assignment
# with Unknown Processing Time

**Shinji Ito**
NEC Corporation, RIKEN AIP
i-shinji@nec.com

**Daisuke Hatano**
RIKEN AIP
daisuke.hatano@riken.jp

**Hanna Sumita**
Tokyo Institute of Technology
sumita@c.titech.ac.jp

**Kei Takemura**
NEC Corporation
kei_takemura@nec.com

**Takuro Fukunaga**
Chuo University
fukunaga.07s@g.chuo-u.ac.jp

**Naonori Kakimura**
Keio University
kakimura@math.keio.ac.jp

**Ken-ichi Kawarabayashi**
National Institute of Informatics, The University of Tokyo
k_keniti@nii.ac.jp

## Abstract

This study considers a novel problem setting, referred to as *bandit task assignment*, that incorporates the processing time of each task in the bandit setting. In this problem setting, a player sequentially chooses a set of tasks to start so that the set of processing tasks satisfies a given combinatorial constraint. The reward and processing time for each task follow unknown distributions, values of which are revealed only after the task has been completed. The problem generalizes the stochastic combinatorial semi-bandit problem and the budget-constrained bandit problem. For this problem setting, we propose an algorithm based on upper confidence bounds (UCB) combined with a phased-update approach. The proposed algorithm admits a gap-dependent regret upper bound of $O(MN(1/\Delta)\log T)$ and a gap-free regret upper bound of $\tilde{O}(\sqrt{MNT})$, where $N$ is the number of the tasks, $M$ is the maximum number of tasks run at the same time, $T$ is the time horizon, and $\Delta$ is the gap between expected per-round rewards of the optimal and best suboptimal sets of tasks. These regret bounds nearly match lower bounds.

## 1 Introduction

This paper introduces a new model of sequential decision-making that we refer to as the *bandit task assignment* problem. The goal of this model is to determine which tasks to perform sequentially so that the total reward will be maximized, while estimating the distribution of rewards and processing time for each task. For each round $t = 1, 2, \ldots, T$, we choose a set $A_t$ of tasks to start. Each task $i$ in $A_t$ will be completed in the $(t + c_{ti})$-th round, and we then obtain the reward $r_{ti}$. The processing time $c_{ti}$ and the reward $r_{ti}$ follow unknown distributions independently for all $t$, and they are observed only when the task has been completed, i.e., only bandit feedback is available. In all rounds, the set of processing tasks is required to satisfy a given combinatorial constraint. That is, in each round, the player can start a set of new tasks that, together with the tasks still in progress, satisfies the given

37th Conference on Neural Information Processing Systems (NeurIPS 2023).

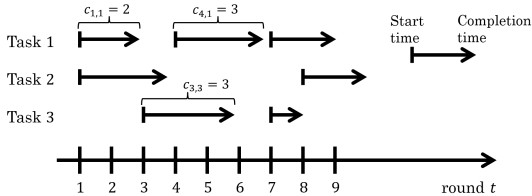

Figure 1: An example of a feasible task assignment.

constraint. The performance of the algorithm is evaluated by means of regret $R_T$, which is defined as the difference between the cumulative rewards obtained with an optimal policy and with the algorithm. Our model allows an arbitrary constraint for a set of tasks that is expressed by $\mathcal{A} \subseteq \{0,1\}^N$, where $N$ is the number of tasks. Each vector $a = (a_1, a_2, \ldots, a_N)^\top \in \mathcal{A}$ is interpreted as a feasible set of tasks $A = \{i \in [N] \mid a_i = 1\}$, where $[N] := \{1, 2, \ldots, N\}$.

Figure 1 illustrates an example of a feasible task assignment for the case of $\mathcal{A} = \{a \in \{0,1\}^3 \mid \|a\|_1 \leq 2\}$, in which we choose which tasks to start from three candidates so that the number of processing tasks does not exceed two. Each arrow in this figure represents the interval of time to process a task. In this example, at round 2, two tasks (Tasks 1 and 2) are still processing, and hence, we cannot start any new task. At the beginning of round 3, we observe that Task 1 is completed, and it is then possible to start one new task. We note that the processing time $c_{ti}$ and the reword $r_{ti}$ of task $i$ stated at time $t$ are revealed only at the completion time $(t + c_{ti})$.

The model in this paper is a common generalization of stochastic combinatorial semi-bandit problems [25, 24, 34] and budget-constrained bandit problems [20, 15, 12, 37]. In fact, if the processing time $c_{ti}$ is always equal to 1 for any task $i$, the player can choose any set of $\mathcal{A}$ in every round and can observe the feedback immediately afterward, which coincides with the stochastic combinatorial semi-bandit problem. On the other hand, if we suppose that $\mathcal{A} = \{a \in \{0,1\}^N \mid \|a\|_1 \leq 1\}$, i.e., if only a single task can be performed in every round, the problem corresponds to stochastic $N$-armed bandits with a budget constraint [20, 15]. Indeed, by regarding $c_{ti}$ as the cost for choosing arm $i$, we can interpret the model as a problem with the budget constraint that the cumulative cost is at most $T$.

We note that our problem setting is similar to *blocking bandits* [8], in which each chosen arm is *blocked*, i.e., is unavailable for a certain period of time after it is chosen. However, they are different in the constraints on the actions in each round. For example, consider $\mathcal{A} = \{a \in \{0,1\}^N \mid \|a\|_1 \leq K\}$. Then, in the bandit task assignment problem, we cannot start a new task when $K$ tasks are already running. On the other hand, in the blocking bandit problem, the player can choose any $K$ arms that are not blocked. Blocking bandits have been extended to contextual bandit models [9], semi-bandit models [3, 28], and adversarial models [10], as well.

Applications of our model include the following examples:

**Example 1** (Matching constraint). We consider the problem of assigning $L$ jobs to $K$ workers. Let $V_1$ and $V_2$ be the sets of workers and jobs, respectively. Let $E \subseteq V_1 \times V_2$ be a set of pairs of a worker and a job such that $(u, v) \in E$ if and only if the worker $u$ is qualified to do the job $v$. In each round, we choose a subset $M$ of $E$ so that no two elements in $M$ share the same worker or job, that is, $M$ must satisfy the matching constraint for the bipartite graph $G = (V_1 \cup V_2, E)$. Each job has processing time to complete, and, when a job is assigned to a worker, the job-worker pair is occupied until the job is completed. Thus the occupied job-worker pairs have to be included in $M$. We assume that the time required to complete a job and the quality of the results (rewards) follow unknown distributions that depend on the job-worker pair. This problem can be regarded as a special case of the bandit task assignment problem with $N = |E|$. Choosing $a_t \in \mathcal{A}$ corresponds to assigning a job $v$ to a worker $u$ for each $e = (u, v)$ such that $a_{te} = 1$, and the worker $u$ and the job $v$ will then be occupied and unavailable until the task is completed. It would be worth mentioning that the matching constraint is often the subject of consideration in the context of combinatorial semi-bandits [18, 30].

**Example 2** (Knapsack constraint). Consider the situation where we want to perform tasks (e.g., of data processing) under resource constraints (e.g., computational resources such as CPUs, GPUs, and memories). We assume that the processing time of each task varies stochastically, and is not revealed until it is completed. Formally, we are given $N$ tasks and $K$ resources. Assume that each task $i \in [N]$ consumes $b_{ki} \in \mathbb{R}_{\geq 0}$ amount of resource $k \in [K]$ per unit time, and that the total

amount of resource $k$ available in unit time is $f_k$. Then the player can choose a set of tasks from $\mathcal{A} = \{a \in \{0,1\}^N \mid b_k^\top a \leq f_k \quad (k \in [K])\}$, where $b_k = (b_{k1}, b_{k2}, \ldots, b_{kN})^\top \in \mathbb{R}_{\geq 0}^N$ for each resource $k$. Note that the set of *all running tasks* must satisfy the constraint given by $\mathcal{A}$, which means that we cannot use resources occupied by tasks in progress and we are restricted to choose a set of tasks that can be executed with remaining resources. We note that the class of problems with a single resource ($K = 1$) includes the budget-constrained multi-armed bandit [20, 15] as a special case.

**Example 3** (Matroid constraint [35, 17]). If $\mathcal{A}$ consists of indicator vectors for independent sets of a matroid with the ground set $[N]$, we call this a matroid constraint. Matroid constraints capture a variety of problem settings of practical importance, including the situation in which the number of tasks that can be executed simultaneously is limited to at most $K$, i.e., $\mathcal{A} = \{a \in \{0,1\}^N \mid \|a\|_1 \leq K\}$. This special case is called a uniform matroid.

## 1.1 Contribution

Our main contribution is to present an algorithm that achieves a regret bound of $O(\frac{C_{\mathrm{u}}}{C_l^2} MN(1/\Delta + C_{\mathrm{u}}) \ln T)$, where $C_{\mathrm{u}}$ and $C_l$ are upper and lower bounds for the processing times $c_{ti}$, $M$ is the maximum of $\|a\|_1$ for $a \in \mathcal{A}$, and $\Delta > 0$ is the gap between expected per-round rewards for the optimal and the best suboptimal actions. The proposed algorithm also enjoys a gap-free regret bound of $O(\frac{1}{C_l}\sqrt{C_{\mathrm{u}} MNT \ln T})$. Our contributions include a nearly tight lower bound as well, i.e., we show that any algorithm suffers a regret at least $\Omega(\frac{1}{C_l}\sqrt{C_{\mathrm{u}} MNT})$. This indicates that the proposed algorithm achieves nearly optimal regret bounds.

To design our algorithm, we first show that a nearly optimal policy can be characterized by expected per-round rewards for tasks (Proposition 2.2 in Section 2). This implies that, given estimation for expected per-round rewards and a linear optimization oracle for $\mathcal{A}$, we can compute asymptotically optimal actions. This is in contrast to the blocking bandits, in which it is computationally hard to maximize the total reward even if all the distributions are known, as presented, e.g., in Corollary 3.2 by Basu et al. [8]. That is why, for the blocking bandits, existing algorithms can achieve only approximate regret bounds, i.e., the performance is only guaranteed to be (close to) a constant-factor approximation for the optimal strategy.

Proposition 2.2 suggests us to balance exploration and exploitation with the aid of upper confidence bounds (UCB) on the expected per-round reward for each task, which can be computed efficiently. Such a UCB-based approach is similar to `CombUCB1` for combinatorial semi-bandits [25]. `CombUCB1`, however, does not directly apply to our problem setting due to the stochastically variable processing time. In the bandit task assignment problem, the set of tasks that the player can start in each round changes depending on the set of tasks still in progress, which makes the decision of the player restricted. To address this issue, we introduce a *phased-update* approach, in which we divide the entirety of the rounds $[T] = \{1, 2, \ldots, T\}$ into *segments* of appropriate length. At the beginning of each segment $s$, we compute a set of tasks $A_s'$ based on the UCB for expected per-round rewards. We note that this can be done independently of the state of running tasks. Then, in the $s$-th segment, we basically continue to perform the same set of tasks $A_s'$.

One main technical difficulty is how to determine the length $l_s$ of the segment $s$. At the start of a new segment $s$, tasks in $A_s'$ cannot be executed until all tasks performed in the previous segment have been completed, i.e., every switching point between segments causes *idle time* with non-zero probability. Therefore, the larger the number of segments (i.e., the shorter the length of each segment), the greater the loss due to such waiting time. In particular, if the segment length is $O(C)$, we cannot avoid a linear regret of $\Omega(T/C)$ due to idle time. On the other hand, setting too long segments also has a negative impact on performance as the longer the length of each segment, the less frequently the set $A_s'$ will be updated, which may degrade the efficiency of exploration and exploitation. To address this dilemma, this study designs a way of determining the segment length $l_s$ by which these potential losses will be well balanced and the overall regret will be bounded as desired.

Another nontrivial technique in the proposed algorithm is to employ *Bernstein-type confidence bounds* [4, 27] for the processing time. Thanks to this, we can construct a tight UCB estimator for the expected per-round reward for each task $i$, with a width depending on the mean $\bar{c}_i$ and variance $\sigma_i^2$ of the processing time. This is essential to achieve the nearly optimal $\tilde{O}(\frac{1}{C_l}\sqrt{C_{\mathrm{u}} MNT})$-regret bound. In fact, as we mention in Remark 4.2, an algorithm with standard confidence bounds will lead to regret upper bounds with additional $C_{\mathrm{u}}/C_l$ factors, which do not match the lower bound.

**Summary of contributions**  The contributions of this paper are summarized in the following four points: (i) We propose a novel problem setting of the bandit task assignment, which incorporates simultaneously combinatorial constraints among tasks, (unknown) processing time of each task, and the exploration-exploitation trade-off. The model includes various practical applications as described in Examples 1, 2 and 3. (ii) We show that a nearly optimal policy can be characterized with expected per-round rewards, which can be computed efficiently given only a linear optimization oracle. This contrasts with the computational difficulty in blocking bandits. (iii) We provide an algorithm with nearly optimal regret bounds. We handle the difficulties arising from the combinatorial constraints together with processing times by the phased-update approach. (iv) We present a regret lower bound that matches to the regret bound of the proposed algorithm, ignoring logarithmic factors. This means that the proposed algorithm achieves nearly optimal performance in terms of the worst-case analysis.

## 2  Problem Setup

Let $N$ be the number of tasks. The player is given a family of feasible sets of tasks, which is expressed by $\mathcal{A} \subseteq \{0,1\}^N$. Here, each binary vector $a$ in $\{0,1\}^N$ corresponds to a set of tasks, and $a \in \mathcal{A}$ means that the set of tasks $A = \{i \in [N] \mid a_i = 1\}$ can be executed simultaneously. We assume that $\mathcal{A}$ is closed under inclusion, i.e., $a \in \mathcal{A}$ and $b \leq a$ together imply $b \in \mathcal{A}$. Note here that, for any vectors $a = (a_1, \ldots, a_N)^\top, b = (b_1, \ldots, b_N)^\top \in \mathbb{R}^N$, the notation of $b \leq a$ means that $b_i \leq a_i$ holds for all $i \in \{1, \ldots, N\}$. We denote $M = \max_{a \in \mathcal{A}} \|a\|_1$. In each round $t$, the player will choose a set of tasks $a_t$ from $\mathcal{A}$. The chosen set has to exclude all the processing tasks not yet completed. More precisely, $a_t$ is constrained to satisfy $a_t + b_t \in \mathcal{A}$, where $b_t$ is the set of tasks still in progress at the $t$-th round (see (1) below). Each task $i$ with $a_{ti} = 1$ is then started and will be completed at the beginning of the $(t + c_{ti})$-th round, which yields the reward of $r_{ti}$. The reward $r_{ti}$ and the processing time $c_{ti}$ will be revealed only after the task is completed. We assume that $r_{ti}$ and $c_{ti}$ are restricted so that $r_{ti} \in [0,1]$ and $c_{ti} \in \{C_l, C_l + 1, \ldots, C_u\}$, where $C_l$ and $C_u$ are integers satisfying $1 \leq C_l \leq C_u$. We also assume that $((r_{ti}, c_{ti}))_{i=1}^N$ follows an unknown distribution $\mathcal{D}$ over $([0,1] \times \{C_l, \ldots, C_u\})^N$, independently for $t = 1, 2, \ldots, T$. Note that $r_{ti}$ and $c_{ti}$ may be dependent. From the above problem definition, a family $\mathcal{A}_t$ of task sets available at the $t$-th round is expressed as follows:

$$b_{ti} = \sum_{s=1}^{t-1} a_{si} \mathbf{1}[s + c_{si} > t] \quad (i \in [N]), \quad \mathcal{A}_t = \{a \in \mathcal{A} \mid a + b_t \in \mathcal{A}\}, \tag{1}$$

where the vector $b_t = (b_{ti})_{i=1}^N$ corresponds to the set of uncompleted tasks that starts before the $t$-th round. The goal of the player is to maximize the sum of rewards earned by the $T$-th round, which can be expressed as $\sum_{t=1}^T r_t^\top a_t$.

*Remark* 2.1.  Stochastic combinatorial semi-bandits [25] and the multi-armed bandit problem with a budget constraint [15] are special cases of the bandit task assignment problem. In fact, if $c_{ti} = 1$ for all $t$ and $i$, the problem corresponds to a combinatorial semi-bandit problem with action set $\mathcal{A}$. In addition, if $\mathcal{A} = \{a \in \{0,1\}^N \mid \|a\|_1 \leq 1\}$, the problem corresponds to an $N$-armed bandit with budget $B = T$ and costs $\{c_{ti}\}$.

In the problem, we assume that we are able to solve linear optimization over $\mathcal{A}$ efficiently. More precisely, we suppose that we are given access to an *offline linear optimization oracle* that returns a solution $\hat{a} \in \arg\max_{a \in \mathcal{A}} q^\top a$ for any input $q \in \mathbb{R}^N_{\geq 0}$. Note here that, for the problems with matching constraints or with matroid constraints (Examples 1 and 3, respectively), such an oracle can be implemented in polynomial time [23, 17]. For Example 2, linear optimization over $\mathcal{A}$ is NP-hard, but can be solved practically fast in many cases, with the aid of dynamic programming or integer programming algorithms.

We define the *regret* $R_T$ to be the expectation of the difference between the total rewards obtained with the optimal *proper policy* and those obtained with the algorithm by the $T$-th round, where we call a policy for choosing $a_t$ *proper* if $a_t$ satisfies the constraint defined by (1) and each $a_t$ depends on information observed by the beginning of $t$-th round, but not on $((r_s, c_s))_{s=t}^T$. Any proper policy can be expressed as sequences of mapping $\pi = (\pi_t)_{t=1,2,\ldots}$, where each $\pi_t$ maps from the history $h_t$, which consists of chosen actions $(a_s)_{s=1}^{t-1}$ and feedback obtained by the beginning of the $t$-round, to

action $a_t$ in $\mathcal{A}$. Let $\Pi$ denote the set of all proper policies. The regret can then be expressed as

$$R_T = \max_{\pi^* \in \Pi} \mathbf{E}\left[\sum_{t=1}^{T} r_t^\top \pi_t^*(h_t^*)\right] - \mathbf{E}\left[\sum_{t=1}^{T} r_t^\top a_t\right],$$

where $(h_t^*)_{t=1,2,\dots}$ denotes histories generated by the policy $\pi^*$. Intuitively, the total reward obtained with the optimal proper policy corresponds to the maximum performance achieved by an agent with unlimited computational resources who knows the distribution $\mathcal{D}$ of processing times and rewards. Note that the regret $R_T$ is defined on the basis of tasks *started by* the $T$-th round. Although it may seem more natural to define the regret by the rewards for tasks *completed by* the $T$-th round, the difference between these two alternatives is only $O(M/C_1)$, which is almost negligible as it is independent of $T$.

We can see that the performance of the optimal proper policy (and therefore also regret) can be characterized by using the expected per-round rewards. We define $\bar{r}_i$ and $\bar{c}_i$ to be the expectation of $r_{ti}$ and $c_{ti}$, respectively. We also denote $q_i = \bar{r}_i / \bar{c}_i$. Define vectors $\bar{r} \in [0,1]^N, \bar{c} \in [C_1, C_u]^N$ and $q \in [0, 1/C_1]^N$ by $\bar{r} = (\bar{r}_1, \bar{r}_2, \dots, \bar{r}_N)^\top, \bar{c} = (\bar{c}_1, \bar{c}_2, \dots, \bar{c}_N)^\top$ and $q = (q_1, q_2, \dots, q_N)^\top = (\bar{r}_1/\bar{c}_1, \bar{r}_2/\bar{c}_2, \dots, \bar{r}_N/\bar{c}_N)^\top$. Note that $q_i$ corresponds to the expected per-round reward for running each task $i$. Then, the expected total reward for an optimal policy can be bounded as follows:

**Proposition 2.2.** *For any proper policy $\pi = (\pi_t)_{t=1,2,\dots} \in \Pi$, we have $\mathbf{E}\left[\sum_{t=1}^{T} r_t^\top \pi_t(h_t)\right] \leq (T + C_u) \max_{a \in \mathcal{A}} \{q^\top a\}$, where $(h_t)_{t=1}^{T}$ denote histories generated by $\pi$.*

Proposition 2.2 can be considered as a generalization of Lemma 1 by Ding et al. [15]. From this proposition, the regret can be bounded as follows:

$$R_T \leq (T + C_u) \max_{a \in \mathcal{A}} q^\top a - \mathbf{E}\left[\sum_{t=1}^{T} r_t^\top a_t\right] \leq \mathbf{E}\left[\sum_{t=1}^{T} (q^\top a^* - r_t^\top a_t)\right] + \frac{C_u M}{C_1}, \qquad (2)$$

where we denote $a^* \in \arg\max_{a \in \mathcal{A}} q^\top a$ and the last inequality follows from the facts that $q_i \leq 1/C_1$ holds for any $i$ and that $\|a^*\|_1 \leq M$.

## 3 Algorithm

The proposed algorithm employs a phased-update approach, in which we update a policy for each phase, and each phase continues to select the same set of tasks.

The procedure starts with the *initialization phase* (or the zeroth phase), in which we execute each task $B = \Theta(\frac{C_u}{C_1} \ln T)$ times. This is required to ensure that the UCB estimators $\hat{q}_i(t)$ given in (3) are close enough to the true expected reward $q_i$. Let $t_1 \geq 1$ denote the round this initialization terminates, which is at most $O(C_u N B)$. As the regret per round is at most $\frac{M}{C_1}$, the total regret in the initialization phase is at most $O(\frac{C_u M N B}{C_1}) = O(\frac{C_u^2}{C_1^2} M N \ln T)$, which turns out to be negligible in the overall regret upper bound.

The $s$-th phase ($s \in \{1, 2, \dots\}$) consists of a segment of rounds with length $l_s$, which implies that the first round $t_s$ of the $s$-th phase is expressed as $t_s = \sum_{u=1}^{s-1} l_u + t_1$. At the beginning of each phase $s$, we compute $a_s' \in \mathcal{A}$ and the length $l_s$ of the $s$-th phase, for which the details will be described later. Then, in the subsequent $l_s$ rounds, we continue to choose the set $A_s' = \{i \in [N] \mid a_{si}' = 1\}$. More precisely, in the $t$-th round, we choose $a_t = a_s' - b_t$ if $b_t \leq a_s'$ and choose $a_t = 0$ otherwise, where we recall $b_t$ is the set of tasks in progress at the $t$-th round (see (1)). Since $a_s' \in \mathcal{A}$, $a_t$ clearly belongs to $\mathcal{A}_t$. In this procedure, intuitively, we repeatedly restart each task $i$ in $A_s'$ just after $i$ is completed, in the $s$-th phase. Note here that all tasks executed in the $(s-1)$-st phase are completed by the $(t_s + C_u)$-th round as the processing times are at most $C_u$, which implies that $b_t \leq a_s'$ holds for any $t \in [t_s + C_u, t_{s+1} - 1]$. Hence, in any round $t \in [t_s + C_u, t_{s+1} - 1]$, it holds that $b_t + a_t = a_s'$, which implies all tasks in $A_s'$ are running in the phase. Furthermore, for each task $i \in A_s'$, the number of times the task $i$ is completed in the $s$-th phase will be at least $\lfloor (l_s - C_u)/C_u \rfloor (\geq l_s/C_u - 2)$ and at most $l_s/C_1 + 1$.

We next describe how to compute $a_s'$. We use *upper confidence bounds (UCB)* for $q_i = \bar{r}_i/\bar{c}_i$, the expected reward per round. In our definition of the UCB, we use the empirical means $\hat{r}_i(t)$ and $\hat{c}_i(t)$

of rewards and costs, the empirical variance $V_i^c(t)$ of costs, and the number $T_i(t)$ of times that the task $i$ has been completed before the $t$-th round. We define the UCB $\hat{q}_i(t)$ on $q_i$ by

$$\hat{q}_i(t) = \frac{\min\{1, \hat{r}_i(t) + d_i^r(t)\}}{\max\{C_\mathrm{l}, \hat{c}_i(t) - d_i^c(t)\}}, \quad \text{where } d_i^r(t) = \sqrt{\frac{1.5 \ln t}{T_i(t)}},$$

$$d_i^c(t) = \sqrt{\frac{3 V_i^c(t) \ln t}{T_i(t)}} + \frac{9(C_\mathrm{u} - C_\mathrm{l}) \ln t}{T_i(t)}. \tag{3}$$

We would like to note this definition includes an *empirical Bernstein bound* $d_i^c(t)$ [4] for the processing time $c_{ti}$, which will turn out to be essential for tight regret upper bounds, as mentioned in Remark 4.2. We then have

$$\Pr\left[|\hat{r}_i(t) - \bar{r}_i| \geq d_i^r(t)\right] \leq \frac{2}{t^2}, \quad \Pr\left[|\hat{c}_i(t) - \bar{c}_i| \geq d_i^c(t)\right] \leq \frac{4}{t^2}. \tag{4}$$

These can be shown from Azuma–Hoeffding inequality and empirical Bernstein's inequality [4], of which details can be found in Section A.4 of the appendix. Hence, with probability $1 - 6/t^2$, we have $q_i \leq \hat{q}_i(t)$. At the beginning of each phase $s$, i.e., in the $t_s$-th round, we calculate these UCBs $\hat{q}(t_s) = (\hat{q}_1(t_s), \hat{q}_2(t_s), \ldots, \hat{q}_N(t_s))^\top$, and then call the linear optimization oracle to find $a'_s \in \arg\max_{a \in \mathcal{A}} \hat{q}(t_s)^\top a$.

We set the length $l_s$ of the $s$-th phase on the basis of $\min_{i \in A'_s} T_i(t_s)$, where we denote $A'_s = \{i \in [N] \mid a'_{si} = 1\}$. We set $l_s = C_\mathrm{l} \min_{i \in A'_s} T_i(t_s) + 2C_\mathrm{u}$. In the analysis, we assume that $B$ is given by $B = \lceil 90 \frac{C_\mathrm{u}}{C_\mathrm{l}} \ln T \rceil$. We then have

$$T_i(t_s) \geq 90 C_\mathrm{u} \ln T / C_\mathrm{l}, \quad l_s \geq 90 C_\mathrm{u} \ln T, \quad 90 C_\mathrm{u} N \ln T \leq t_1 = O\left(C_\mathrm{u}^2 N \ln T / C_\mathrm{l}\right), \tag{5}$$

which will be used at several points in our analysis. These conditions and the construction of $l_s$ lead to the bound on $T_i(t_s)$ as in the lemma below, which will be used in the regret analysis for the proposed algorithm.

**Lemma 3.1.** *For any $s \geq 1$ and $i \in A'_s$, we have $l_s \leq 2C_\mathrm{u}(T_i(t_{s+1}) - T_i(t_s))$ and $T_i(t_{s+1}) \leq 4T_i(t_s)$.*

All omitted proofs are given in the appendix. We also have the following bound on $t_s$:

**Lemma 3.2.** *For any $s \geq 1$, there exists $i \in A'_s$ such that $T_i(t_{s+1}) \geq (1 + C_\mathrm{l}/C_\mathrm{u})T_i(t_s)$. Consequently, we have $t_s \geq C_\mathrm{l}(1 + C_\mathrm{l}/C_\mathrm{u})^{s/N-2}$ for any $s \geq 1$.*

From Lemma 3.2, we have $\ln t_s \geq \ln C_\mathrm{l} + \left(\frac{s}{N} - 2\right) \ln \left(1 + \frac{C_\mathrm{l}}{C_\mathrm{u}}\right) \geq \ln C_\mathrm{l} + \left(\frac{s}{N} - 2\right) \frac{C_\mathrm{l}}{2C_\mathrm{u}}$, where the last inequality follows from the fact that $\ln(1 + x) \geq x/2$ for any $x \in [0, 1]$. This implies that we have $s \leq N \left(\frac{2C_\mathrm{u}}{C_\mathrm{l}} \ln t_s + 2\right)$. Hence, the number of phases, denoted by $S$, is bounded by $S \leq N \left(\frac{2C_\mathrm{u}}{C_\mathrm{l}} \ln T + 2\right) + 1 = O\left(\frac{C_\mathrm{u}}{C_\mathrm{l}} N \ln T\right)$, since the last phase contains $T$.

The overall procedure of the proposed algorithm is summarized in Algorithm 1, which consists of arithmetic operations and offline linear optimization over $\mathcal{A}$. The number of arithmetic operations in each round is bounded by $O(N)$ since the update of each parameter in Step 7–10 of Algorithm 1 can be performed in $O(1)$-arithmetic operations. Furthermore the number of calls to the offline linear optimization oracle is at most $S = O\left(\frac{C_\mathrm{u}}{C_\mathrm{l}} N \ln T\right)$. In fact, the offline linear optimization oracle is called only at the beginning of each phase. This means that Algorithm 1 is more efficient than standard UCB algorithms for combinatorial semi-bandits algorithms [24, 25] that require $O(T)$ calls to the oracle. The space complexity, other than that required for the offline optimization oracle, is $O(N)$ since the algorithm works by maintaining $O(N)$ parameters.

## 4 Regret Analysis

We denote $a^* \in \arg\max_{a \in \mathcal{A}} q^\top a$. Let $A^* = \{i \in [N] \mid a_i^* = 1\}$ and $\tilde{A} = [N] \setminus A^*$. We define *suboptimality gaps* $\Delta_a$, $\Delta_i$ and $\Delta_\mathrm{min}$, respectively, by

$$\Delta_a = q^\top a^* - q^\top a \quad (a \in \mathcal{A}), \quad \Delta_i = \min_{a \in \mathcal{A}: a_i = 1, \Delta_a > 0} \Delta_a \quad (i \in [N]), \quad \Delta_\mathrm{min} = \min_{i \in \tilde{A}} \Delta_i. \tag{6}$$

**Algorithm 1** Phased-update UCB algorithm for bandit task assignment

---

**Require:** $\mathcal{A} \subseteq \{0,1\}^N$: a family of feasible task sets. $C_\mathrm{l}, C_\mathrm{u}$: lower and upper bounds for the processing time. $B$: length of initialization phase. Offline linear optimization oracle for $\mathcal{A}$.

1: For each $i \in [N]$, execute the $i$'s task $B$ times, and let $t_1$ be the round it completed. Set $T_i(t_1) = B$.
2: **for** $s = 1, 2, \ldots,$ **do**
3:     Set $\hat{q}_i(t_s)$ by (3).
4:     Call the offline linear optimization oracle to find $a'_s \in \arg\max_{a\in\mathcal{A}} \hat{q}(t_s)^\top a$, and set $A'_s = \{i \in [N] \mid a'_{si} = 1\}$.
5:     Set $l_s = C_\mathrm{l} \min_{i \in A'_s} T_i(t_s) + 2C_\mathrm{u}$ and $t_{s+1} = t_s + l_s$.
6:     **for** $t = t_s, t_s + 1, \ldots, t_{s+1} - 1$ **do**
7:         If $b_t \leq a'_s$, output $a_t = a'_s - b_t$, where $b_t$ is given by (1). Otherwise, output $a_t = 0$.
8:         **for** $i = 1, 2, \ldots, N$ **do**
9:             If task $i$ is completed (i.e., $b_{ti} = 0$) and $c_{t',i}$ and $r_{t',i}$ are observed ($t'$ is the last round the task $i$ is started), update $\bar{c}_i(t)$, $\bar{r}_i(t)$ and $V_i^c(t)$ by using $c_{t',i}$ and $r_{t',i}$. Set $T_i(t+1) = T_i(t) + 1$.
10:         **end for**
11:     **end for**
12: **end for**

---

We also denote the variance of the processing time $c_{ti}$ by $\sigma_i^2 = \mathbf{E}[(c_{ti} - \bar{c}_i)^2]$. We note that $\sigma_i^2 \leq \mathbf{E}[(c_{ti} - C_\mathrm{l})^2] \leq (C_\mathrm{u} - C_\mathrm{l})\,\mathbf{E}[c_{ti} - C_\mathrm{l}] = (C_\mathrm{u} - C_\mathrm{l})(\bar{c}_i - C_\mathrm{l})$.

The goal of this section is to show the following:

**Theorem 4.1.** *The regret for Algorithm 1 is bounded as*

$$R_T \leq O\left( M \sum_{i \in \tilde{A}} \frac{\tilde{C}_i \ln T}{\Delta_i} + \frac{C_\mathrm{u}^2}{C_\mathrm{l}^2} NM \ln T \right) \leq O\left( \left( \frac{1}{\Delta_{\min}} + C_\mathrm{u} \right) \frac{C_\mathrm{u} NM \ln T}{C_\mathrm{l}^2} \right), \quad (7)$$

*where* $\tilde{C}_i \leq O\left( \frac{1}{\bar{c}_i}\left(1 + \frac{1}{\bar{c}_i^2}\left(\sigma_i^2 + \frac{(C_\mathrm{u}-C_\mathrm{l})^2 C_\mathrm{l}}{C_\mathrm{u}}\right)\right)\right) \leq O\left(\frac{1}{\bar{c}_i}\left(1 + \frac{C_\mathrm{u}-C_\mathrm{l}}{\bar{c}_i}\right)\right) \leq O\left(\frac{C_\mathrm{u}}{\bar{c}_i C_\mathrm{l}}\right)$. *Furthermore, for any distribution $\mathcal{D}$, the regret is bounded as* $R_T = O\left( \frac{1}{C_\mathrm{l}}\sqrt{C_\mathrm{u} NMT \ln T} + \frac{C_\mathrm{u}^2}{C_\mathrm{l}^2} NM \ln T \right)$.

The specific definition of $\tilde{C}_i$ is given in (19) in the appendix.

*Remark* 4.2. Bernstein-type confidence bounds used in (3) are essential to achieve the nearly optimal regret bound of $R_T = O\left( \frac{1}{C_\mathrm{l}}\sqrt{C_\mathrm{u} NMT \ln T} \right)$. In fact, if we employ a standard confidence bound for $\bar{c}_i$ instead of the Bernstein-type one given in (3), the parameter $\sigma_i^2$ in the definition of $\tilde{C}_i$ will be replaced with $(C_\mathrm{u} - C_\mathrm{l})^2$, which leads a suboptimal regret bound of $R_T = O\left( \sqrt{\frac{C_\mathrm{u}^2}{C_\mathrm{l}^3} T \ln T} \right)$.

From Proposition 2.2, the regret can be bounded as follows:

$$R_T \leq \mathbf{E}\left[ \sum_{t=1}^{T} (q^\top a^* - r_t^\top a_t) \right] + \frac{C_\mathrm{u}}{C_\mathrm{l}} M \leq \frac{M}{C_\mathrm{l}} \mathbf{E}\,[t_1] + R_T^{(1)} + R_T^{(2)} + \frac{C_\mathrm{u}}{C_\mathrm{l}} M, \quad (8)$$

where we define

$$R_T^{(1)} = \mathbf{E}\left[ \sum_{s=1}^{S} \sum_{t=t_s}^{t_{s+1}-1} (q^\top a^* - q^\top a'_s) \right], \quad R_T^{(2)} = \mathbf{E}\left[ \sum_{s=1}^{S} \sum_{t=t_s}^{t_{s+1}-1} (q^\top a'_s - r_t^\top a_t) \right]. \quad (9)$$

In these definitions of $R_T^{(1)}$ and $R_T^{(2)}$, the index $S$ represents the phase that includes the $T$-th round, and we define $t_{S+1} = T + 1$ exceptionally, for notational simplicity.

Let us next show the upper bounds for $R_T^{(1)}$ and $R_T^{(2)}$ separately. We will show that

$$R_T^{(1)} = O\left(M \sum_{i \in \tilde{A}} \frac{\tilde{C}_i \ln T}{\Delta_i} + \frac{C_u^2}{C_l^2} NM\right), \quad R_T^{(2)} = O\left(\frac{C_u^2}{C_l^2} NM \ln T\right). \tag{10}$$

These, together with (8) and (5), prove the gap-dependent regret bound (7) in Theorem 4.1.

We here discuss an upper bound on $R_T^{(1)}$ while the analysis of $R_T^{(2)}$ is given in the supplementary material. We denote $\tilde{A}_s = A_s' \setminus A^*$. Define $d_i(t)$ by $d_i(t) = \sqrt{\frac{\tilde{C}_i}{\tilde{c}_i} \frac{\ln t}{T_i(t)}}$, where $\tilde{C}_i$ is defined by (19) in the appendix so that $\tilde{C}_i = \Theta\left(\frac{1}{\tilde{c}_i}\left(1 + \frac{1}{\tilde{c}_i^2}\left(\sigma_i^2 + \frac{(C_u - C_l)^2 C_l}{C_u}\right)\right)\right)$. We then have $\hat{q}_i(t) - q_i(t) \leq d_i(t)$ with high probability. In fact, we can show the following lemma using concentration inequalities, of which proof can be found in the appendix.

**Lemma 4.3.** *For any $s$, with a probability at least $1 - 6N/t_s^2$, we have $\Delta_{a_s'} \leq \sum_{i \in \tilde{A}_s} d_i(t_s)$. Consequently, we have $R_T^{(1)} \leq \mathbf{E}[\hat{R}_T] + O\left(\frac{MN}{C_l t_1}\right)$, where we define $\hat{R}_T = \sum_{s=1}^{S} l_s \Delta_{a_s'} \mathbf{1}[\mathcal{F}_s]$ with $\mathcal{F}_s = \left\{\Delta_{a_s'} \leq \sum_{i \in \tilde{A}_s} d_i(t_s), \ \Delta_{a_s'} > 0\right\}$.*

We can show an upper bound on $\hat{R}_T$ in a way similar to what Kveton et al. [25] did, as follows:

**Lemma 4.4.** *$\hat{R}_T$ defined in Lemma 4.3 is bounded as $\mathbf{E}[\hat{R}_T] = O\left(M \sum_{i \in \tilde{A}} \frac{\tilde{C}_i \ln T}{\Delta_i}\right)$.*

This bound on $\hat{R}_T$ together with Lemma 4.3 and (5) leads to the first part of (10).

### 4.1 Gap-Free Regret Bound

We can obtain the gap-free regret bound of $R_T = \tilde{O}(\sqrt{\frac{C_u}{C_l^2} NMT})$ in Theorem 4.1 by modifying the analysis of $\hat{R}_T$. For any $\epsilon > 0$, $\hat{R}_T$ can be bounded as $\hat{R}_T \leq \sum_{s=1}^{S} \left(l_s \Delta_{a_s'} \mathbf{1}[\mathcal{F}_s, \Delta_{a_s'} > \epsilon] + l_s \Delta_{a_s'} \mathbf{1}[\mathcal{F}_s, \Delta_{a_s'} \leq \epsilon]\right) = O\left(\frac{C_u}{C_l^2} MN \frac{\ln T}{\epsilon} + \epsilon T\right)$, where the last inequality follows from the same argument for showing Lemma 4.4. By setting $\epsilon = \sqrt{\frac{C_u MN \ln T}{C_l^2 T}}$, we obtain $\hat{R}_T = O\left(\frac{1}{C_l}\sqrt{C_u MNT \ln T}\right)$. From this, (8), the second part of (10), and Lemma 4.3, we obtain the gap-free regret bound presented in Theorem 4.1. This completes the proof of Theorem 4.1.

### 4.2 Regret Lower Bound

As shown in Theorem 4.1, the proposed algorithm achieves $R_T = O(\frac{1}{C_l}\sqrt{C_u NMT})$. This is tight up to a logarithmic factor in $T$. In fact, we have the following lower bound:

**Theorem 4.5.** *For any $N, M, T$ such that $N \geq M$ and $T = \Omega(C_u)$, there exists a problem instance for which any algorithm suffers regret of $R_T = \Omega(\frac{1}{C_l} \min\{\sqrt{C_u NMT}, MT\})$.*

In the proof of this theorem, we first focus on the case of $M = 1$. If we consider the case of $c_{ti} = C_l$, our model is equivalent to the $N$-armed bandit with time horizon $T' = \Theta(T/C_l)$, which leads to a lower bound of $\Omega(\sqrt{NT'}) = \Omega(\sqrt{NT/C_l})$ with the aid of the lower bound for multi-armed bandit problems given in Theorem 5.1 by Auer et al. [6]. When considering the case of $r_{ti} = 1$ and $c_{ti} \in \{C_l, C_u\}$, we can show a regret lower bound of $\Omega(\frac{C_u - C_l}{C_l C_u}\sqrt{C_u NT})$, by using the proof technique by Badanidiyuru et al. [7]. Combining these two bounds, we obtain an $\Omega(\frac{1}{C_l}\sqrt{C_u NT})$-lower bound for the case of $M = 1$. From this result and the technique used in Proposition 2 by Kveton et al. [25], we have an $\Omega(\frac{1}{C_l}\sqrt{C_u MNT})$-lower bound for $M > 1$.

## 5 Numerical Experiment

We conducted small experiments to evaluate the practical performance of the proposed algorithm using a synthetic dataset. We set the parameters for the dataset as follows: $\mathcal{A} = \{A \subseteq [N] \mid$

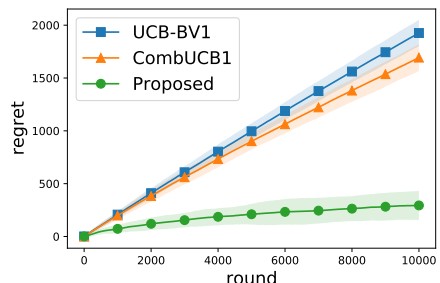

Figure 2: Instance with small $\Delta$.

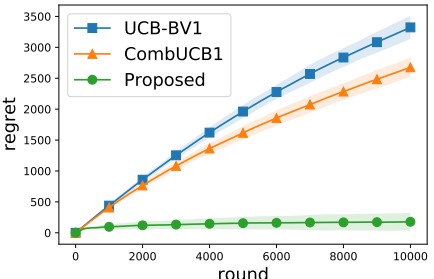

Figure 3: Instance with large $\Delta$.

$|A| \leq M\}$ with $(N, M) = (4, 2)$, $T = 10000$, $C_u = 6$, $C_l = 1$, $\bar{r} = (0.5, 0.5, 0.5, 0.5)$. For the expected processing times, we set $\bar{c} = (1.5, 1.5, 2.0, 2.0)$ as a problem instance with small $\Delta$ or $\bar{c} = (1.5, 1.5, 5.0, 5.0)$ as a problem instance with large $\Delta$. Each $c_{ti}$ and $r_{ti}$ are generated so that $c_{ti} - 1$ follows binomial distributions over $\{0, 1, \ldots, C_u - 1\}$ and $r_{ti}$ follows Bernoulli distributions over $\{0, 1\}$. For comparison, we have added the results of applying UCB-BV1 by Ding et al. [15] and CombUCB1 by Kveton et al. [24]. More detailed information on the experimental setup is provided in the supplementary material.

Figures 2 and 3 show the empirical means of regret computed by 100 repetitions of independent experiments. We depict 2-standard deviations of the empirical regret by the shaded areas. These results imply that the proposed algorithm performs better experimentally than other algorithms. We note that UCB-BV1 and CombUCB1 cannot avoid a linear regret as they choose a set in $\mathcal{A}$ and wait for them all to be completed, which causes idling time with non-zero probability every time. It is also worth mentioning that the empirical performance of the proposed algorithm is much better than Theorem 4.1 predicts. In fact, under the parameter settings of these experiments, the values in Theorem 4.1 can be approximated as: $\frac{1}{C_l}\sqrt{C_u NMT \ln T} \approx 2100$ and $\frac{C_u}{C_l} NM \ln T \approx 2650$.

## 6 Related Work

One of the most relevant studies to this work would be those on *combinatorial semi-bandits*, in which a family $F \subseteq 2^{[N]}$ of subsets of arms $[N]$ is given and the player sequentially chooses $A_t \in F$ and then observes the obtained rewards $r_{ti}$ for each $i \in A_t$. Studies on combinatorial semi-bandits are classified into those on stochastic models [25, 24, 34], in which $r_t$ are i.i.d. for $t$, and those on adversarial models [33, 5], in which $\{r_t\}$ are arbitrary sequences that may be chosen in an adversarial manner. As noted in the introduction, the stochastic combinatorial semi-bandits are special cases of bandit task assignment. The stochastic combinatorial semi-bandits are known to admit sublinear regret. Typical approaches to achieve sublinear regret include upper confidence bounds (UCB)-type algorithms [25, 24] and Thompson sampling algorithms [34].

Our problem is also relevant to bandit problems with budget constraints. Ding et al. [15] consider a stochastic multi-armed bandit (MAB) problem with a budget constraint, in which the player observes the reward $r_{ti_t}$ and the cost $c_{ti_t}$ for the chosen arm $i_t$ in each round $t$. The goal of the player is to maximize the cumulative rewards earned before cumulative costs reach a given total budget amount. This can be seen as a special case of our problem. As shown in Ding et al. [15], in the budget-constrained MAB problem, the reward of the optimal policy can be characterized by the expected reward-to-cost. This fact is generalized to our setting in Proposition 2.2, which will be used in our regret analysis. It should be mentioned that the budget-constrained MAB problem has been generalized to bandits with multiple plays [37], which differs from our model, as problems with processing times cannot be reduced to budget-constrained models in general. In addition to their work, there are various generalizations and variants, such as contextual bandit models [36], models with general distributions [11, 12], and models with multiple budgets [2, 31, 7].

While delayed feedback models [13, 38, 3] are similar to our model in that rewards earned will be revealed in later rounds, there are some essential differences. For example, in our model, the selected arm is occupied during the processing time and cannot be selected, while in the delayed feedback model, the arm can be selected while waiting for feedback. The former model can handle situations

where the arm corresponds to a labor or computing resource, as in the examples shown in Examples 1 and 2.

Phased-update or lazy-update approaches were incorporated in other bandit problems and online learning problems such as a phased version of UCB for MAB (e.g., Exercise 7.5 of the book [26]) and the rarely switching OFUL algorithm for linear stochastic bandits [1]. The motivation for using phased-update approaches is primarily to reduce the computational cost [1, 22], to minimize switching cost [2, 21], or to address batched settings [29, 19]. To our knowledge, our proposed algorithm is the first algorithm that applies such a phased-update approach to combinatorial semi-bandits. It allows us to reduce the number of oracle calls, and more importantly, to achieve the nearly tight regret bounds for the bandit task assignment. Our study reveals new usefulness of a phased-update approach.

Another line of research related to our problem is the *online matching* or *online allocation problems with reusable resources* [14, 16, 32]. The problem is formalized with a bipartite graph, and we choose edges for sequentially arriving vertices. The model is similar to ours in that each chosen edge will be unavailable for a certain period of time that follows a probability distribution. However, it is different in that the bipartite graph is unknown at first and all the distributions are given. Moreover, those papers adopt competitive ratio analyses rather than regret analyses, i.e., they evaluate the performance compared to a stronger agent who knows the sequence of arriving vertices. Thus, the existing papers are incomparable with our work.

As mentioned in the introduction, the problem setting of *blocking bandit* [8] appears to be similar to ours while the difficulties are significantly different. Considering unconstrained situations can highlight the differences in the problem. In the task assignment problem with no constraints, the optimal strategy will be trivial, in which we make each task always processing, i.e., in each round, we start every task that is completed at that round. On the other hand, as the reviewer commented, blocking bandits only allow us to play a single action in each round, which makes the optimal strategy nontrivial. In fact, finding the optimal policy can be an NP-hard problem even if the true distributions of $r$ and $c$ are given. This is an illustrative example of why bandit task assignment can be easier than blocking bandits.

## 7    Conclusion and limitation

In this paper, we introduced a new problem setting that we referred to as bandit task assignment, for which we proposed an algorithm with regret bounds. A limitation of this work is that we need to know the upper and lower bounds $C_u, C_l$ on the processing times, which may not be possible to assume in some real-world applications. To remove this assumption, it would be necessary to modify the framework of the problem. One possible modification would be to allow the player to suspend processing tasks. In this modified problem setup, even in cases where $C_u$ is incorrectly underestimated, it will be possible to immediately abort the task and correct $C_u$ when the underestimation is detected. We expect that this strategy and the appropriate parameter update rules for $C_u$ and $C_l$ allow us to remove the assumption of prior knowledge about processing times. Another future direction is to extend the problem to a decentralized setting, in which multiple agents collaborate to make task assignment decisions. We believe that such an extension of the problem provides insight into real-world applications where multiple entities perform task assignments, such as crowdsourcing and federated learning.

## Acknowledgments

We deeply appreciate many comprehensive comments and feedback by anonymous reviewers. HS was supported by JSPS KAKENHI Grant Numbers JP17K12646, JP21K17708, and JP21H03397, Japan. TF was supported by JSPS KAKENHI Grant Numbers JP20H05965, JP21K11759, and JP21H03397, Japan. NK was supported by JSPS KAKENHI Grant Numbers JP22H05001, JP20H05795, and JP21H03397, Japan. KK was supported by JSPS KAKENHI Grant Numbers JP22H05001 and JP20A402, Japan.

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

# A Omitted Proofs

**Lemma A.1** (Empirical bernstein inequalities, [4]). *Let $X_1, X_2, \ldots, X_n \in [C_\mathrm{l}, C_\mathrm{u}]$ be a sequence of i.i.d. random variables with mean $\mu$ and variance $\sigma^2$. Let $\hat{\mu}_n = \sum_{i=1}^n X_i / n$ and $V_n = \sum_{i=1}^n (X_i - \hat{\mu}_n)^2 / n$. For any $\delta \in (0, 1)$, with probability $1 - 4\delta$, we have*

$$|\hat{\mu}_n - \mu| \leq \sqrt{\frac{V_n}{n} \ln \frac{1}{\delta}} + \frac{3(C_\mathrm{u} - C_\mathrm{l})}{n} \ln \frac{1}{\delta} \leq \sqrt{\frac{\sigma^2}{n} \ln \frac{1}{\delta}} + \frac{5(C_\mathrm{u} - C_\mathrm{l})}{n} \ln \frac{1}{\delta}. \tag{11}$$

*Proof.* Theorem 1 by Audibert et al. [4] implies that the first inequality holds with probability $1 - 3\delta$. The second inequality follows from Bernstein's inequality for $V_t$:

$$\Pr\left[ V_t \geq \sigma^2 + \sqrt{\frac{2(C_\mathrm{u} - C_\mathrm{l})^2 \sigma^2}{n} \ln \frac{1}{\delta}} + \frac{2(C_\mathrm{u} - C_\mathrm{l})^2}{3n} \ln \frac{1}{\delta} \right] \leq \delta. \tag{12}$$

From this, with probability $1 - \delta$, we have

$$\sqrt{\frac{V_n}{n} \ln \frac{1}{\delta}} \leq \sqrt{\frac{1}{n}\left( \sigma^2 + \sqrt{\frac{2(C_\mathrm{u} - C_\mathrm{l})^2 \sigma^2}{n} \ln \frac{1}{\delta}} + \frac{2(C_\mathrm{u} - C_\mathrm{l})^2}{3n} \ln \frac{1}{\delta} \right) \ln \frac{1}{\delta}}$$

$$= \sqrt{\frac{\sigma^2}{n} \ln \frac{1}{\delta}} \cdot \sqrt{1 + \sqrt{\frac{2(C_\mathrm{u} - C_\mathrm{l})^2}{\sigma^2 n} \ln \frac{1}{\delta}} + \frac{2(C_\mathrm{u} - C_\mathrm{l})^2}{3\sigma^2 n} \ln \frac{1}{\delta}}$$

$$\leq \sqrt{\frac{\sigma^2}{n} \ln \frac{1}{\delta}} \cdot \left( 1 + 2\sqrt{\frac{(C_\mathrm{u} - C_\mathrm{l})^2}{\sigma^2 n} \ln \frac{1}{\delta}} \right) \leq \sqrt{\frac{\sigma^2}{n} \ln \frac{1}{\delta}} + \frac{2(C_\mathrm{u} - C_\mathrm{l})}{n} \ln \frac{1}{\delta},$$

where the second inequality follows from $\sqrt{1 + x + y} \leq \sqrt{1 + x} + \sqrt{y} \leq 1 + x/2 + \sqrt{y}$ that holds for any $x, y \geq 0$. This implies that the second inequality of (11) holds with probability $1 - \delta$. $\square$

## A.1 Proof of Proposition 2.2

*Proof.* For $t = 1, 2, \ldots, T$, denote $a_t = \pi_t(h_t)$ and define $b_t$ by (1). For $t > T$, let $a_t = 0$ for the notational simplicity. Then, as $r_t$ is independent of $a_t$, we have

$$\mathbf{E}\left[ \sum_{t=1}^T r_t^\top a_t \right] = \mathbf{E}\left[ \sum_{t=1}^T \bar{r}^\top a_t \right] = \bar{r}^\top \mathbf{E}\left[ \sum_{t=1}^T a_t \right]. \tag{13}$$

From (1), we have $\mathbf{E}\left[ \sum_{t=1}^{T+C_\mathrm{u}} (a_{ti} + b_{ti}) \right] = \mathbf{E}\left[ \sum_{t=1}^{T+C_\mathrm{u}} \left( a_{ti} + \sum_{s=1}^{t-1} a_{si} \mathbf{1}[s + c_{si} \geq t] \right) \right] = \mathbf{E}\left[ \sum_{t=1}^T a_{ti} c_{ti} \right] = \bar{c}_i \mathbf{E}\left[ \sum_{t=1}^T a_{ti} \right]$ for any $i$, where the last equality follows from the fact that $c_t$ is independent of $a_t$. Hence, we have $\mathbf{E}\left[ \sum_{t=1}^{T+C_\mathrm{u}} (a_t + b_t) \right] = D \mathbf{E}\left[ \sum_{t=1}^T a_t \right]$, where $D \in \mathbb{R}_{>0}^{N \times N}$ denote the diagonal matrix with entries $\bar{c}_1, \bar{c}_2, \ldots, \bar{c}_N$. As we have $a_t + b_t \in \mathcal{A}$, this implies that $\frac{1}{T+C_\mathrm{u}} D \mathbf{E}\left[ \sum_{t=1}^T a_t \right] = \mathbf{E}\left[ \frac{1}{T+C_\mathrm{u}} \sum_{t=1}^{T+C_\mathrm{u}} (a_t + b_t) \right]$ is in the convex hull of $\mathcal{A}$. We hence have $\bar{r}^\top \mathbf{E}\left[ \sum_{t=1}^T a_t \right] = (T + C_\mathrm{u})(D^{-1}\bar{r})^\top \left( \frac{1}{T+C_\mathrm{u}} D \mathbf{E}\left[ \sum_{t=1}^T a_t \right] \right) \leq (T + C_\mathrm{u}) \max_{a \in \mathcal{A}} \left\{ q^\top a \right\}$, where the last inequality follows from $D^{-1}\bar{r} = q$ and the fact that $\frac{1}{T+C_\mathrm{u}} D \mathbf{E}\left[ \sum_{t=1}^T a_t \right]$ is in the convex hull of $\mathcal{A}$. Combining this with (13), we obtain the bound in Proposition 2.2. $\square$

## A.2 Proof of Lemma 3.1

*Proof.* From (5), we have $\min_{i \in A_s'} T_i(t_s) \geq C_\mathrm{u}/C_\mathrm{l}$, which implies $l_s = C_\mathrm{l} \min_{i \in A_s'} T_i(t_s) + 2C_\mathrm{u} \geq 2C_\mathrm{u}$. From this, for all $i \in A_s'$, we have $T_i(t_{s+1}) - T_i(t_s) \geq (l_s - 2C_\mathrm{u})/C_\mathrm{u} \geq l_s/(2C_\mathrm{u})$. This means the first inequality in Lemma 3.1 holds. Further, we have

$$T_i(t_{s+1}) - T_i(t_s) \leq (l_s + 1)/C_\mathrm{l} \leq (C_\mathrm{l} \min_{i \in A_s'} T_i(t_s) + C_\mathrm{u} + 1)/C_\mathrm{l}$$

$$= \min_{i \in A_s'} T_i(t_s) + C_\mathrm{u}/C_\mathrm{l} + 1/C_\mathrm{l} \leq 3 \min_{i \in A_s'} T_i(t_s),$$

which implies that the second inequality in Lemma 3.1 holds. $\square$

## A.3 Proof of Lemma 3.2

*Proof.* Let $i^* \in A_s'$ be a task such that $T_{i^*}(s) = \min_{i \in A_s'} T_i(t_s)$. We then have $l_s = C_1 T_{i^*}(s) + 2C_\mathrm{u}$. As the task $i^*$ is completed at least $(l_s - 2C_\mathrm{u})/C_\mathrm{u} = (C_1/C_\mathrm{u}) T_{i^*}(s)$ times within the $s$-th phase, we have $T_{i^*}(s+1) - T_{i^*}(s) \geq (C_1/C_\mathrm{u}) T_{i^*}(s)$, which implies $T_{i^*}(s+1) \geq (1 + C_1/C_\mathrm{u}) T_{i^*}(s)$. Further, if $i \in A_{s'}'$ for some $s' < s$, we have $T_i'(s) \geq C_\mathrm{u}/C_1$ as shown in the proof of Lemma 3.1. Hence, if $s \geq NK + 1$ for some positive integer $K$, there exists $i \in [N]$ such that $T_{t_i}(s) \geq (C_\mathrm{u}/C_1)(1 + C_1/C_\mathrm{u})^{K-1}$. From this, as $t_s \geq C_1 T_i(t_s)$ holds for any $i \in [N]$, we have $t_s \geq C_1(1 + C_1/C_\mathrm{u})^{s/N-2}$. $\qquad\square$

## A.4 Proof of Lemma 4.3

Let us first show that $\Delta_{a_s'} \leq \sum_{i \in \tilde{A}_s} d_i(t_s)$ holds with a probability of at least $1 - 6N/t_s^2$. Fix $t$ arbitrarily.

For any $i$ and let $\tau(1) < \tau(2) < \cdots$ represent all the indices of rounds at which the $i$-th task is started. Define the sequence $E_1, E_2, \ldots$ by $E_j = \sum_{s=1}^j (r_i(\tau(s)) - \bar{r}_i)$. Then, since $E_j$ is a sum of $j$ i.i.d. random variables, the Azuma–Hoeffding inequality implies $\Pr[|E_j| \geq \sqrt{1.5j \cdot \ln t}] \leq \frac{2}{t}$ for any $t$ and $j$. We hence have

$$\Pr\left[|\hat{r}_i(t) - \bar{r}_i| \geq d_i^r(t)\right] = \Pr\left[|E_{T_i(t)}| \geq \sqrt{1.5 T_i(t) \cdot \ln t}\right]$$

$$\leq \Pr\left[\exists j \in [t], \quad |E_j| \geq \sqrt{1.5j \cdot \ln t}\right] \leq \sum_{j=1}^t \Pr\left[|E_j| \geq \sqrt{1.5j \cdot \ln t}\right] \leq \sum_{j=1}^t \frac{2}{t^3} \leq \frac{2}{t^2}. \quad (14)$$

which means that the first part of (4) holds. Similarly, by considering $E_j = \sum_{s=1}^j (c_i(\tau(s)) - \bar{c}_i)$. it can be shown from Lemma A.1 that

$$\Pr\left[|\hat{c}_i(t) - \bar{c}_i| \leq d_i^c(t) \leq \sqrt{\frac{3\sigma_i^2 \ln t}{T_i(t)}} + \frac{15(C_\mathrm{u} - C_1) \ln t}{T_i(t)}\right] \geq 1 - \frac{4}{t^2}. \quad (15)$$

We further have

$$\sqrt{\frac{3\sigma_i^2 \ln t}{T_i(t)}} + \frac{15(C_\mathrm{u} - C_1) \ln t}{T_i(t)} \leq \left(\sqrt{3\sigma_i^2} + 15(C_\mathrm{u} - C_1)\sqrt{\frac{C_1}{90C_\mathrm{u}}}\right)\sqrt{\frac{\ln t}{T_i(t)}}$$

$$\leq \sqrt{3\sigma_i^2}\sqrt{\frac{C_1}{90C_\mathrm{u}}} + 15(C_\mathrm{u} - C_1)\frac{C_1}{90C_\mathrm{u}} \leq \sqrt{(C_\mathrm{u} - C_1)(\bar{c}_i - C_1)}\sqrt{\frac{C_1}{30C_\mathrm{u}}} + (C_\mathrm{u} - C_1)\frac{C_1}{6C_\mathrm{u}}$$

$$\leq \frac{\sqrt{(\bar{c}_i - C_1)C_1}}{\sqrt{30}} + \frac{C_1}{6} \leq \frac{\bar{c}_i}{4}, \quad (16)$$

where the first and second inequalities follow from the lower bound on $T_i(t_s)$ in (5), and the third and last inequalities follow from $C_1 \leq \bar{c}_i \leq C_\mathrm{u}$. Hence, with probability $1 - 6N/t^2$,

$$\hat{q}_i(t) = \frac{\min\{1, \hat{r}_i(t) + d_i^r(t)\}}{\max\{C_1, \hat{c}_i(t) - d_i^c(t)\}} \geq \frac{\bar{r}_i}{\bar{c}_i} = q_i. \quad (17)$$

As (16) implies $\bar{c}_i - 2d_i^c(t) \geq \bar{c}_i/2$, we have

$$\hat{q}_i(t) = \frac{\min\{1, \hat{r}_i(t) + d_i^r(t)\}}{\max\{C_1, \hat{c}_i(t) - d_i^c(t)\}} \leq \frac{\min\{1, \bar{r}_i + 2d_i^r(t)\}}{\max\{C_1, \bar{c}_i - 2d_i^c(t)\}} =: \frac{\tilde{r}_i}{\tilde{c}_i}$$

$$= \frac{\bar{r}_i}{\bar{c}_i} + \frac{\tilde{r}_i - \bar{r}_i}{\bar{c}_i} + \frac{\tilde{r}_i(\bar{c}_i - \tilde{c}_i)}{\bar{c}_i \tilde{c}_i} \leq q_i + \frac{2d_i^r(t)}{\bar{c}_i} + \frac{2d_i^c(t)}{\bar{c}_i \tilde{c}_i} \leq q_i + \frac{2d_i^r(t)}{\bar{c}_i} + \frac{4d_i^c(t)}{\bar{c}_i^2}$$

$$\leq q_i + \left(\frac{2\sqrt{1.5}}{\bar{c}_i} + \frac{4}{\bar{c}_i^2}\left(\sqrt{3\sigma_i^2} + 15(C_\mathrm{u} - C_1)\sqrt{\frac{C_1}{90C_\mathrm{u}}}\right)\right)\sqrt{\frac{\ln t_s}{T_i(t_s)}}$$

$$= q_i + \sqrt{\frac{\tilde{C}_i}{\bar{c}_i}\frac{\ln t_s}{T_i(t_s)}} = q_i + d_i(t_s), \quad (18)$$

where we denote $\tilde{r}_i = \min\{1, \bar{r}_i + 2d_i^r(t)\}$ and $\tilde{c}_i = \max\{C_1, \bar{c}_i - 2d_i^c(t)\} \geq \bar{c}_i/2$. We here defined $\tilde{C}_i$ by

$$\tilde{C}_i = \frac{1}{\bar{c}_i}\left(2\sqrt{1.5} + \frac{4}{\bar{c}_i}\left(\sqrt{3\sigma_i^2} + 15(C_u - C_1)\sqrt{\frac{C_1}{90C_u}}\right)\right)^2. \tag{19}$$

We then have

$$\begin{aligned}
\Delta_{a'_s} &= \sum_{i \in A^* \setminus A'_s} q_i - \sum_{i \in A'_s \setminus A^*} q_i \leq \sum_{i \in A^* \setminus A'_s} \hat{q}_i(t_s) - \sum_{i \in A'_s \setminus A^*} q_i \\
&\leq \sum_{i \in A'_s \setminus A^*} \hat{q}_i(t_s) - \sum_{i \in A'_s \setminus A^*} q_i \leq \sum_{i \in A'_s \setminus A^*} d_i(t_s) = \sum_{i \in \tilde{A}_s} d_i(t_s),
\end{aligned} \tag{20}$$

where the first and last inequalities follow from (17) and (18), respectively, and the second inequality follows from the definition of $a'_s$. Indeed, $a'_s \in \arg\max_{a \in \mathcal{A}} \hat{q}(t_s)^\top a$ implies the following inequality: $\sum_{i \in A^* \setminus A'_s} \hat{q}_i(t_s) - \sum_{i \in A'_s \setminus A^*} \hat{q}_i(t_s) = \hat{q}(t_s)^\top a^* - \hat{q}(t_s)^\top a'_s \leq 0$.

Let us next show the bound on $R_T^{(1)}$. We decompose $R_T^{(1)}$ as follows:

$$R_T^{(1)} = \mathbf{E}\left[\sum_{s=1}^S l_s \Delta_{a'_s}\right] = \mathbf{E}\left[\sum_{s=1}^S l_s \Delta_{a'_s} \mathbf{1}[\mathcal{F}_s] + \sum_{s=1}^S l_s \Delta_{a'_s} \mathbf{1}[\bar{\mathcal{F}}_s]\right] = \mathbf{E}\left[\hat{R}_T + \sum_{s=1}^S l_s \Delta_{a'_s} \mathbf{1}[\bar{\mathcal{F}}_s]\right], \tag{21}$$

where $\bar{\mathcal{F}}_s$ denotes the complement of the event $\mathcal{F}_s$. The second part of RHS can be bounded as

$$\begin{aligned}
\mathbf{E}\left[\sum_{s=1}^S l_s \Delta_{a'_s} \mathbf{1}[\bar{\mathcal{F}}_s]\right] &= \mathbf{E}\left[\sum_{s=1}^S l_s \Delta_{a'_s} \mathbf{1}\left[\Delta_{a'_s} > \sum_{i \in \tilde{A}_s} d_i(t_s)\right]\right] \\
&\leq \mathbf{E}\left[\sum_{s=1}^S l_s \frac{M}{C_1} \frac{6N}{t_s^2}\right] = \frac{6MN}{C_1} \mathbf{E}\left[\sum_{s=1}^S \frac{l_s}{t_s^2}\right]
\end{aligned} \tag{22}$$

where the first inequality follows from the fact that (20) holds with a probability of at least $1 - 6N/t_s^2$ and the inequality of $\Delta_{a'_s} \leq M/C_1$. We have

$$\frac{l_s}{t_s^2} = \frac{t_{s+1} - t_s}{t_s^2} = \frac{t_{s+1} - t_s}{t_s^2} \frac{t_{s+1}}{t_s} = \left(\frac{1}{t_s} - \frac{1}{t_{s+1}}\right)\frac{t_{s+1}}{t_s} \leq 5\left(\frac{1}{t_s} - \frac{1}{t_{s+1}}\right),$$

where the last inequality follows from

$$\frac{t_{s+1}}{t_s} = 1 + \frac{l_s}{t_s} \leq 1 + \frac{1}{t_s}\left(C_1 \max_{i \in [N]} T'_i(s) + 2C_u\right) \leq 1 + \frac{3C_1}{t_s} \max_{i \in [N]} T'_i(s) \leq 1 + \frac{3C_1}{t_s}\left(\frac{t_s}{C_1} + 1\right) \leq 5.$$

Hence, we have

$$\sum_{s=1}^S \frac{l_s}{t_s^2} \leq 5\sum_{s=1}^S \left(\frac{1}{t_s} - \frac{1}{t_{s+1}}\right) \leq 5\frac{1}{t_1}. \tag{23}$$

Combining (21), (22) and (23), we obtain the bound on $R_T^{(1)}$ in Lemma 4.3.

## A.5 Proof of Lemma A.2

The following lemma can be shown in a way similar to that of Kveton et al. [25, Lemma 3].

**Lemma A.2.** *Let $\{\alpha_k\}$ and $\{\beta_k\}$ be positive real sequences for which $1 = \beta_0 > \beta_1 > \cdots > \beta_k > \cdots > 0$, $\alpha_1 > \alpha_2 > \cdots > \alpha_k > \cdots > 0$, $\lim_{k \to \infty} \alpha_k = \lim_{k \to \infty} \beta_k = 0$, and $\sum_{k=1}^\infty \frac{\beta_{k-1} - \beta_k}{\sqrt{\alpha_k}} \leq 1$. Suppose that $\mathcal{F}_s$ occurs. There then exists $k$ such that $S_{s,k} = \{i \in \tilde{A}_s \mid \Delta_{a'_s} \leq M\sqrt{\alpha_k}d_i(t_s)\}$ satisfies $|S_{s,k}| \geq \beta_k M$.*

*Proof.* Let us show the claim via proof by contradiction. Suppose that $|S_{s,k}| < \beta_k M$ holds for all $k$. We then have

$$\sum_{i \in \tilde{A}_s} d_i(t_s) < \sum_{k=1}^{\infty} \sum_{i \in S_{s,k-1} \setminus S_{s,k}} \frac{\Delta_{a'_s}}{\tilde{C} M \sqrt{\alpha_k}} = \frac{\Delta_{a'_s}}{\tilde{C} M} \sum_{k=1}^{\infty} \frac{|S_{s,k-1}| - |S_{s,k}|}{\sqrt{\alpha_k}}$$

$$= \frac{\Delta_{a'_s}}{\tilde{C} M} \left( \frac{|S_{s,0}|}{\sqrt{\alpha_1}} + \sum_{k=1}^{\infty} |S_k| \left( \frac{1}{\sqrt{\alpha_{k+1}}} - \frac{1}{\sqrt{\alpha_k}} \right) \right)$$

$$< \frac{\Delta_{a'_s}}{\tilde{C}} \left( \frac{\beta_0}{\sqrt{\alpha_1}} + \sum_{k=1}^{\infty} \beta_k \left( \frac{1}{\sqrt{\alpha_{k+1}}} - \frac{1}{\sqrt{\alpha_k}} \right) \right)$$

$$= \frac{\Delta_{a'_s}}{\tilde{C}} \sum_{k=1}^{\infty} \frac{\beta_{k-1} - \beta_k}{\sqrt{\alpha_k}} \le \frac{\Delta_{a'_s}}{\tilde{C}},$$

where the first inequality follows from the definition of $S_{s,k}$, i.e., $\tilde{C} M \sqrt{\alpha_k} d_i(t_s) < \Delta_{a'_s}$ holds for all $i \in [N] \setminus S_{s,k}$, and the second inequality follows from the initial assumption made in this proof. This means that $\mathcal{F}_s$ occurs. Hence, if $\mathcal{F}_s$ occurs, thi initial assumption is false, which means that there exists $k$ such that $|S_{s,k}| \ge \beta_k M$. $\qquad \square$

We will apply Lemma A.2 with $\{\alpha_k\}$ and $\{\beta_k\}$ such that $\sum_{k=1}^{\infty} \frac{\alpha_k}{\beta_k} = O(1)$, which are provided in Appendix A.4 in [25]. From Lemma A.2, if $\mathcal{F}_s$ occurs, $\sum_{i \in \tilde{A}} \mathbf{1}[i \in \tilde{A}_s, \Delta_{a'_s} \le 2\tilde{C} M \sqrt{\alpha_k} d_i(t_s)] \ge \beta_k M$ holds for some $k$, which implies

$$\mathbf{1}[\mathcal{F}_s] \le \sum_{k=1}^{\infty} \frac{1}{\beta_k M} \sum_{i \in \tilde{A}} \mathbf{1}[\mathcal{G}_{s,k,i}], \quad \text{where} \quad \mathcal{G}_{s,k,i} = \left\{ i \in \tilde{A}_s, \Delta_{a'_s} \le M \sqrt{\alpha_k} d_i(t_s) \right\}. \quad (24)$$

*Proof.* Using (24), we can bound $\hat{R}_T$ as

$$\hat{R}_T \le \sum_{i \in \tilde{A}} \sum_{k=1}^{\infty} \frac{1}{\beta_k M} \sum_{s=1}^{S} l_s \Delta_{a'_s} \mathbf{1}[\mathcal{G}_{s,k,i}] \le \sum_{i \in \tilde{A}} \sum_{k=1}^{\infty} \frac{\sqrt{\alpha_k}}{\beta_k} \sum_{s=1}^{S} l_s d_i(t_s) \mathbf{1}[\mathcal{G}_{s,k,i}]$$

$$= \sum_{i \in \tilde{A}} \sum_{k=1}^{\infty} \frac{\sqrt{\alpha_k}}{\beta_k} \sum_{s=1}^{S} l_s \sqrt{\frac{\tilde{C}_i}{\bar{c}_i} \frac{\ln t_s}{T_i(t_s)}} \mathbf{1}[\mathcal{G}_{s,k,i}] \le \sqrt{\ln T} \sum_{i \in \tilde{A}} \sqrt{\frac{\tilde{C}_i}{\bar{c}_i}} \sum_{k=1}^{\infty} \frac{\sqrt{\alpha_k}}{\beta_k} \sum_{s=1}^{S} \frac{l_s \mathbf{1}[\mathcal{G}_{s,k,i}]}{\sqrt{T_i(t_s)}}. \quad (25)$$

Let us evaluate $\sum_{s=1}^{S} \frac{l_s \mathbf{1}[\mathcal{G}_{s,k,i}]}{\sqrt{T_i(t_s)}}$ for some fixed $k$ and $i$. If $\mathcal{G}_{s,k,i}$ occurs, we have $\Delta_i \le \Delta_{a'_s} \le M \sqrt{\alpha_k} d_i(t_s) \le M \sqrt{\frac{\alpha_k \tilde{C}_i \ln T}{\bar{c}_i T_i(t_s)}}$ which implies that $\sqrt{T_i(t_s)} \le M \sqrt{\frac{\alpha_k \tilde{C}_i \ln T}{\bar{c}_i}} \frac{1}{\Delta_i} =: \gamma$. We hence have

$$\sum_{s=1}^{S} \frac{l_s \mathbf{1}[\mathcal{G}_{s,k,i}]}{\sqrt{T_i(t_s)}} \le \sum_{s=1}^{S} \frac{l_s \mathbf{1}[i \in \tilde{A}_s] \mathbf{1}[\sqrt{T_i(t_s)} \le \gamma]}{\sqrt{T_i(t_s)}}. \quad (26)$$

The expectation of the right-hand side can be bounded as follows. From Lemma A.4 in the Appendix, we have

$$\mathbf{E} \left[ \frac{l_s \mathbf{1}[i \in \tilde{A}_s]}{\sqrt{T_i(t_s)}} \right] \le 2\bar{c}_i \mathbf{E} \left[ \frac{T_i(t_{s+1}) - T_i(t_s)}{\sqrt{T_i(t_s)}} \right] \quad (27)$$

By using Lemma 3.1, we obtain $\frac{T_i(t_{s+1}) - T_i(t_s)}{\sqrt{T_i(t_s)}} = 3 \frac{T_i(t_{s+1}) - T_i(t_s)}{\sqrt{4 T_i(t_s)} + \sqrt{T_i(t_s)}} \le 3 \frac{T_i(t_{s+1}) - T_i(t_s)}{\sqrt{T_i(t_{s+1})} + \sqrt{T_i(t_s)}} = 3(\sqrt{T_i(t_{s+1})} - \sqrt{T_i(t_s)})$. Combining this with (26) and (27), we obtain $\mathbf{E} \left[ \sum_{s=1}^{S} \frac{l_s \mathbf{1}[\mathcal{G}_{s,k,i}]}{\sqrt{T_i(t_s)}} \right] \le 6\bar{c}_i \sum_{s=1}^{S} \left( \sqrt{T_i(t_{s+1})} - \sqrt{T_i(t_s)} \right) \mathbf{1} \left[ \sqrt{T_i(t_s)} \le \gamma \right] \le 12\bar{c}_i \gamma = 12 M \frac{\sqrt{\alpha_k \bar{c}_i \tilde{C}_i \ln T}}{\Delta_i}$, where we the last inequality follows from Lemma 3.1. Combining this with (25), we obtain $\mathbf{E} \left[ \hat{R}_T \right] = O \left( M \ln T \cdot \sum_{i \in \tilde{A}} \tilde{C}_i \sum_{k=1}^{\infty} \frac{\alpha_k}{\beta_k} \frac{1}{\Delta_i} \right) = O \left( M \sum_{i \in \tilde{A}} \frac{\tilde{C}_i \ln T}{\Delta_i} \right)$, which completes the proof. $\qquad \square$

## A.6 Proof of the second part of (10) (bounding $\hat{R}_T^{(2)}$)

The second part of (10) can be shown via the following:

**Lemma A.3.** *Given $a'_s$, $t_s$ and $t_{s+1}$, we have* $\mathbf{E}\left[\sum_{t=t_s}^{t_{s+1}-1}(q^\top a'_s - r_t^\top a_t)\right] \leq 3M\frac{C_{\mathrm{u}}}{C_{\mathrm{l}}}$.

*Proof.* For any $s \geq 1$ and task $i \in [N]$, the number of times to start task $i$ duaring the $s$-th phase is at least $(T_i(t_{s+1}) - T_i(t_s) - 1)$. We hence have

$$\mathbf{E}\left[\sum_{t=t_s}^{t_{s+1}-1}r_t^\top a_t\right] \geq \mathbf{E}\left[\sum_{i\in A'_s}\bar{r}_i(T_i(t_{s+1}) - T_i(t_s) - 1)\right] \geq \mathbf{E}\left[\sum_{i\in A'_s}\bar{r}_i\frac{l_s - 3C_{\mathrm{u}}}{\bar{c}_i}\right]$$

$$= \mathbf{E}\left[\sum_{i\in A'_s}q_i(l_s - 3C_{\mathrm{u}})\right] \geq \mathbf{E}\left[l_s\sum_{i\in A'_s}q_i\right] - 3\frac{C_{\mathrm{u}}}{C_{\mathrm{l}}}M = \mathbf{E}\left[l_s \cdot q^\top a'_s\right] - 3\frac{C_{\mathrm{u}}}{C_{\mathrm{l}}}M,$$

where the second inequality follows from Lemma A.4 and the third inequality follows from $q_i = \bar{r}_i/\bar{c}_i \leq 1/C_{\mathrm{l}}$ and $|A'_s| \leq M$. This completes the proof. $\square$

We used the following lemma in the proof of Lemma A.3.

**Lemma A.4.** *For any $s \geq 1$ and $i \in A'_s$, we have*

$$\mathbf{E}[T_i(t_{s+1}) - T_i(t_s)|l_s] \geq \frac{l_s - 2C_{\mathrm{u}}}{\bar{c}_i}. \tag{28}$$

*Consequently, we have*

$$l_s \leq 2\bar{c}_i \cdot \mathbf{E}_s\left[T_i(t_{s+1}) - T_i(t_s)\right], \tag{29}$$

*where $\mathbf{E}_s$ denote the conditional expectation given $l_s$ and $T_i(t_s)$.*

*Proof.* We will show (28) by induction in the value of $l_s \geq 0$. It is clear that (28) hold if $l_s \leq 2C_{\mathrm{u}}$. As the inductive hypothesis, we assume that (28) holds if $l_s \leq m$ for some $m \geq 2C_{\mathrm{u}}$. Then, if $l_s = m + 1 > 2C_{\mathrm{u}}$, at least once, task $i \in A'_s$ is started and completed during the $s$-th phase. Let $c_i$ denote the processing time when task $i$ is first started and completed during the $s$-th phase. We then have

$$\mathbf{E}\left[T_i(t_{s+1}) - T_i(t_s)|l_s = m + 1\right] = \sum_{c=C_{\mathrm{l}}}^{C_{\mathrm{u}}}\Pr[c_i = c]\,\mathbf{E}\left[1 + T_i(t_{s+1}) - T_i(t_s)|l_s = m + 1 - c\right]$$

$$= 1 + \sum_{c=C_{\mathrm{l}}}^{C_{\mathrm{u}}}\Pr[c_i = c]\,\mathbf{E}\left[T_i(t_{s+1}) - T_i(t_s)|l_s = m + 1 - c\right]$$

$$\geq 1 + \sum_{c=C_{\mathrm{l}}}^{C_{\mathrm{u}}}\Pr[c_{t,i} = c]\frac{m + 1 - c - 2C_{\mathrm{u}}}{\bar{c}_i}$$

$$= 1 + \frac{m + 1 - \bar{c}_i - 2C_{\mathrm{u}}}{\bar{c}_i} = \frac{m + 1 - 2C_{\mathrm{u}}}{\bar{c}_i},$$

where the inequality follows from the inductive hypothesis. Hence, (28) holds for $l = m + 1$ as well. By induction, it has been shown that (28) holds. Further, as we have $l_s \geq C_{\mathrm{l}}B + 2C_{\mathrm{u}} \geq 4C_{\mathrm{u}}$, we have $l_s - 2C_{\mathrm{u}} \geq l_s/2$, which implies (29) holds. $\square$

From Lemma A.3, we have $\hat{R}_T^{(2)} \leq 2\frac{C_{\mathrm{u}}}{C_{\mathrm{l}}}SM$. By combining this with $S = O\left(\frac{C_{\mathrm{u}}}{C_{\mathrm{l}}}N\ln T\right)$, which follows from Lemma 3.2, we obtain the second part of (10).

## A.7 Proof of Theorem 4.5

We start from the special case in which $M = 1$, i.e., we assume $\mathcal{A} = \{\{i\} \mid i \in [N]\} \cup \{\emptyset\}$. We will show the following two lower bounds separately:

$$R_T = \Omega\left(\min\left\{\sqrt{\frac{1}{C_l}NT}, \frac{T}{C_l}\right\}\right), \tag{30}$$

$$R_T = \Omega\left(\frac{C_u - C_l}{C_l C_u}\min\left\{\sqrt{C_u NT}, T\right\} - \frac{C_u}{C_l}\right). \tag{31}$$

These two bounds together lead to the regret lower bound of

$$R_T = \Omega\left(\frac{1}{C_l}\min\left\{\sqrt{C_u NT}, T\right\}\right). \tag{32}$$

In fact, if $C_u \le 2C_l$, (30) implies that (32) holds as we have $\frac{C_u}{C_l^2} = \Theta(\frac{1}{C_l})$. On the other hand, if $C_u \ge 2C_l$, (31) implies that (32) holds since we have $\frac{C_u - C_l}{C_l C_u} = \Omega(\frac{1}{C_l})$.

From the lower bound of (32), we can obtain

$$R_T = \Omega\left(\frac{1}{C_l}\min\left\{\sqrt{C_u MNT}, MT\right\}\right) \tag{33}$$

by the technique used in the proof of Proposition 2 by Kveton et al. [25], which provides a regret lower bound for combinatorial semi-bandits. To see this, suppose that $N = MK$ for simplicity. The action set of $K$-path combinatorial semi-bandit (in Proposition 2 by Kveton et al. [25]) is expressed as $\mathcal{A}' = \{A_k \mid k \in [K]\} \cup \{\emptyset\}$, where $A_k = \{M(k-1) + j \mid j \in [M]\}$. Note that this does not satisfy the assumption that the action set is closed under inclusion, which is posed in our problem setting. In order to address this assumption, we set $\mathcal{A} = \{A \subseteq [N] \mid \exists A' \in \mathcal{A}', A \subseteq A'\}$, which is closed under inclusion. Let us consider bandit task assignment problems with action set $\mathcal{A}$, instead of $\mathcal{A}'$. We suppose that, for each $k \in [K]$, $r_{ti}$ and $c_{ti}$ take the same values for all $i \in A_i$, i.e., there exists $r'_{tk}$ and $c'_{tk}$ such that $r_{ti} = r'_{tk}$ and $c_{ti} = c'_{tk}$ for all $i \in A_i$. If $r'\_tk$ and $c'\_tk$ are i.i.d. for different $t$, this is a proper problem instance in our model. (Our model does not assume that $r_{ti}$ and $c_{ti}$ are independent for different tasks $i$ as stated in Section 2). For any action $A_t \in \mathcal{A} \setminus \{\emptyset\}$, let $A'_t \in \mathcal{A}'$ be such that $A_t \subseteq A'_t$. Then, if $(A_t)_{t=1}^T$ is feasible, $(A'_t)_{t=1}^T$ is also feasible, and the reward for the latter is greater than or equal to the former. This means that any algorithm can be converted to one that takes only actions in $\mathcal{A}'$ without sacrificing the performance. Hence, it is sufficient to consider only algorithms that choose actions from $\mathcal{A}'$ in the proof of lower bounds. Therefore, regret lower bounds with action set $\{\{k\} \mid k \in [K]\} \cup \{\emptyset\}$, multiplied with $M$, apply to our problem setting. Hence, the lower bound of (32) implies (33) as well.

The rest of this section is dedicated to the proofs of (30) and (31).

**Proof of** (30)  If $M = 1$ and $c_{ti} = C_l$ for all $t$ and $i$ with probability one, the problem is equivalent to a standard $N$-armed bandit problem with time horizon $T' = \Theta(T/C_l)$. As shown in Theorem 5.1 by Auer et al. [6], any algorithm for the $N$-armed bandit problem suffers $\Omega(\min\{\sqrt{NT'}, T'\})$-regret in the worst case with time horizon $T'$. By substituting $T' = \Theta(T/C_l)$ into this lower bound, we obtain (30).

**Proof of** (31)  Set $p = \frac{C_l}{C_u + C_l}$. Fix an arbitrary algorithm. For some fixed $i^* \in [N]$ and $\epsilon \in [0, p/2]$, define a distribution $\mathcal{D}_{i^*, \epsilon}$ of $c$ and $r$ as follows:

$$c_i = \begin{cases} C_u & \text{with probability} \quad p - \epsilon \cdot \mathbf{1}[i = i^*] \\ C_l & \text{with probability} \quad 1 - p + \epsilon \cdot \mathbf{1}[i = i^*] \end{cases},$$
$$r_i = 1 \quad (i \in [N]).$$

We also denote this distribution with $\epsilon = 0$ by $\mathcal{D}_0$, which does not depend on $i^*$. Denote $\bar{c} = pC_u + (1 - p)C_l = \frac{2C_l C_u}{C_l + C_u} = 2pC_u$. Then the expected processing time can be expressed as

$$\bar{c}_i = \mathbf{E}[c_i] = \bar{c} - \epsilon \cdot (C_u - C_l)\mathbf{1}[i = i^*]. \tag{34}$$

Denote the number of completing the task $i$ during $T$ rounds by $T_i$. Then the expected regret can be expressed as

$$R_T \geq \mathop{\mathbf{E}}_{i^*,\epsilon} \left[ (T - C_{\mathrm{u}})\,(\bar{c} - \epsilon \cdot (C_{\mathrm{u}} - C_{\mathrm{l}}))^{-1} - \sum_{i=1}^{N} T_i \right]$$

$$\geq \frac{T}{\bar{c}}\left(1 + \frac{1}{\bar{c}}\epsilon \cdot (C_{\mathrm{u}} - C_{\mathrm{l}})\right) - \mathop{\mathbf{E}}_{i^*,\epsilon}\left[\sum_{i=1}^{N} T_i\right] - \frac{2C_{\mathrm{u}}}{C_{\mathrm{l}}}, \tag{35}$$

where $\mathbf{E}_{i^*,\epsilon}$ means the expectation for the case in which $c$ and $r$ follow $\mathcal{D}_{i^*,\epsilon}$. As the expected value of the number of rounds to process the task $i$ is expressed as $\mathbf{E}_{i^*,\epsilon}[\bar{c}_i T_i]$, we have

$$\mathop{\mathbf{E}}_{i^*,\epsilon}\left[\sum_{i=1}^{N}\bar{c}_i T_i\right] \leq T.$$

From (34), the left-hand side of this can be expressed as

$$\mathop{\mathbf{E}}_{i^*,\epsilon}\left[\sum_{i=1}^{N}\bar{c}_i T_i\right] = \mathop{\mathbf{E}}_{i^*,\epsilon}\left[\bar{c}\sum_{i=1}^{N} T_i - \epsilon \cdot (C_{\mathrm{u}} - C_{\mathrm{l}})T_{i^*}\right].$$

Combining the above, we obtain

$$\mathop{\mathbf{E}}_{i^*,\epsilon}\left[\sum_{i=1}^{N} T_i\right] \leq \frac{1}{\bar{c}}(T + \epsilon \cdot (C_{\mathrm{u}} - C_{\mathrm{l}})\mathop{\mathbf{E}}_{i^*,\epsilon}[T_{i^*}]). \tag{36}$$

Similarly, we can show that

$$\mathop{\mathbf{E}}_{0}\left[\sum_{i=1}^{N} T_i\right] \leq \frac{T}{\bar{c}},$$

where $\mathbf{E}_0$ means the expectation for the case in which $c$ and $r$ follow $\mathcal{D}_0$. This means that there exists $i^*$ such that

$$\mathop{\mathbf{E}}_{0}[T_{i^*}] \leq \frac{T}{\bar{c}N}$$

We assume that $i^*$ satisfies this in the following.

From (35) and (36), we have

$$R_T \geq \epsilon \cdot \frac{C_{\mathrm{u}} - C_{\mathrm{l}}}{\bar{c}} \cdot \left(\frac{T}{\bar{c}} - \mathop{\mathbf{E}}_{i^*,\epsilon}[T_{i^*}]\right) - \frac{2C_{\mathrm{u}}}{C_{\mathrm{l}}}. \tag{37}$$

By a similar argument to Lemma 6.6 of by Badanidiyuru et al. [7], we can show that

$$\mathop{\mathbf{E}}_{i^*,\epsilon}[T_i^*] \leq \frac{3}{4}\frac{T}{\bar{c} - \epsilon \cdot (C_{\mathrm{u}} - C_{\mathrm{l}})} \quad \text{if} \quad \epsilon \leq \frac{1}{16}\sqrt{\frac{p\bar{c}N}{T}}.$$

Hence, if $\epsilon = \min\left\{\frac{p}{4}, \frac{1}{16}\sqrt{\frac{p\bar{c}N}{T}}\right\}$, we have

$$\frac{3}{4}\frac{T}{\bar{c} - \epsilon \cdot (C_{\mathrm{u}} - C_{\mathrm{l}})} \leq \frac{3}{4}\frac{T}{\bar{c} - pC_{\mathrm{u}}/4} = \frac{3}{4}\frac{T}{\bar{c} - \bar{c}/8} = \frac{6}{7}\frac{T}{\bar{c}}.$$

Combining this with (37), we obtain

$$R_T \geq \frac{1}{7}\epsilon \cdot \frac{C_{\mathrm{u}} - C_{\mathrm{l}}}{\bar{c}}\frac{T}{\bar{c}} - \frac{2C_{\mathrm{u}}}{C_{\mathrm{l}}} = \frac{1}{7}\cdot\frac{C_{\mathrm{u}} - C_{\mathrm{l}}}{\bar{c}}\frac{T}{\bar{c}}\min\left\{\frac{p}{4}, \frac{1}{16}\sqrt{\frac{p\bar{c}N}{T}}\right\} - \frac{2C_{\mathrm{u}}}{C_{\mathrm{l}}}$$

$$= \frac{1}{7}\cdot\frac{C_{\mathrm{u}} - C_{\mathrm{l}}}{2pC_{\mathrm{u}}}\frac{T}{2pC_{\mathrm{u}}}\min\left\{\frac{p}{4}, \frac{1}{16}\sqrt{\frac{2p^2 C_{\mathrm{u}}N}{T}}\right\} - \frac{2C_{\mathrm{u}}}{C_{\mathrm{l}}}$$

$$= \Omega\left(\frac{C_{\mathrm{u}} - C_{\mathrm{l}}}{C_{\mathrm{l}}C_{\mathrm{u}}}\min\left\{T, \sqrt{C_{\mathrm{u}}NT}\right\}\right) - \frac{2C_{\mathrm{u}}}{C_{\mathrm{l}}},$$

which completes the proof of (31).

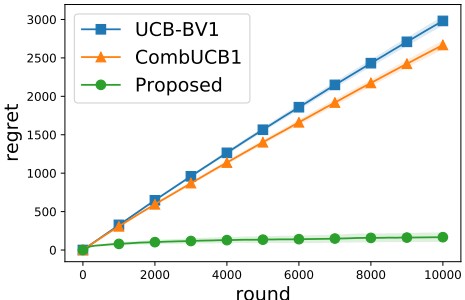

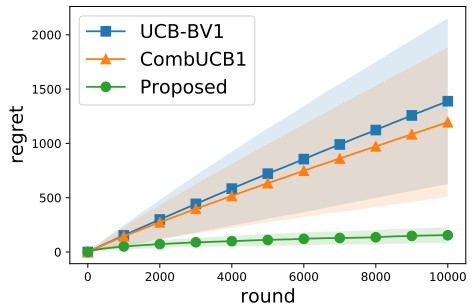

Figure 4: Instance specified by parameters $\bar{c} = (2.19, 4.6, 5.35, 1.42)$ and $\bar{r} = (0.38, 0.43, 0.35, 0.47)$, which are generated from uniform distributions over $[C_l, C_u]$ and $[0, 1]$, respectively.

Figure 5: The average for instances in which the entries of $\bar{c}$ and $\bar{r}$ are generated from uniform distributions over $[C_l, C_u]$ and $[0, 1]$, respectively. The shaded areas represent the standard deviations of the empirical regret.

## B   Notes on experimental setup

In the implementation on UCB-BV1 proposed by Ding et al. [15], by considering each element in $\mathcal{A} = \{A \subseteq [N] \mid |A| = M\}$ as a single arm, we transform an instance of the bandit task assignment into a budget constrained multi-armed bandit problem with $\binom{N}{M}$-arms. To satisfy the constraints, we wait until all tasks in the previously chosen set are completed before deciding the next arm, i.e., the budget consumed by choosing $A_t$ is given as $\max_{i \in A_t} \{c_{ti}\}$.

Similarly, in the implementation of CombUCB1 proposed by Kveton et al. [25], each time a set $A_t$ in $\mathcal{A}$ is selected and it waits until all the tasks in $A_t$ are completed before selecting the next set. However, the way of choosing the set is different from that of UCB-BV1: we maintain the UCB scores for each *task* $i \in [N]$, not for each set $A \in \mathcal{A}$, and then choose a set so that the sum of the UCB scores for tasks in it are maximized, as suggested in the paper by Kveton et al. [25]. Data and source code to reproduce the experimental results in this paper are included in the supplementary materials.

## C   Additional experiments

This section provides the results of numerical experiments performed to evaluate (i) the performance for randomly generated instances and (ii) how the algorithm behaves under different settings, such as varying values of $N$ and $C_u$. Settings of parameters not specified in the captions are the same as in Section 5 and in Appendix B.

Figures 4 and 5 show the results of synthetic experiments for randomly chosen $\bar{r}$ and $\bar{c}$. This experiment confirms that the proposed algorithm works well empirically in randomly generated environments, including cases where the values of $\bar{r}$ are not necessarily constant.

Some results of an empirical evaluation of the algorithm's sensitivity to parameters $N$ and $C_u$ are given in Figures 6, 7, and 8. Figure 6 suggests that the values of the regret scale linearly with respect to $N$, which is consistent with the regret upper bound of (7). Figure 7 shows the computation time for varying problem sizes, which implies that the proposed algorithm is superior in terms of computational complexity. Figure 8 represents how false beliefs about $C_u$ affect the performance of the algorithms. Experimental results suggest that the performance of the proposed algorithm is sensitive to underestimation of $C_u$, while it is robust to overestimation of $C_u$.

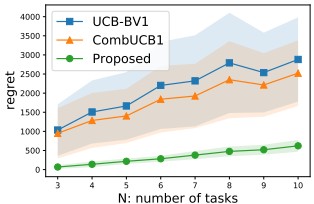

Figure 6: Dependency of the regret on the problem size.

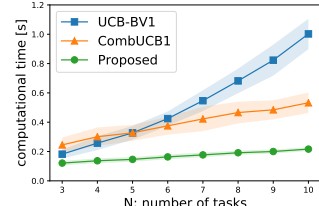

Figure 7: Dependency of the computational time on the problem size.

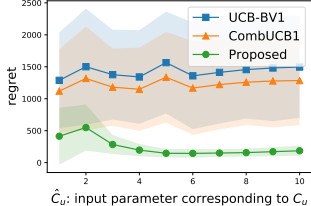

Figure 8: Impact of the mis-specification of $C_{\mathrm{u}}$ on the regret. Note that the true value of $C_{\mathrm{u}}$ is 6.

