# OpenReview forum: "Bandit Task Assignment with Unknown Processing Time"
_NeurIPS.cc/2023/Conference — NeurIPS 2023 poster_

### Official Review · Reviewer_PHPg · 2023-07-03

**Soundness:** 3 good
**Presentation:** 3 good
**Contribution:** 3 good
**Rating:** 6
**Confidence:** 3

**Summary:**

The paper studies the *bandit task assignment* problem.
In this problem, a player chooses tasks sequentially,
each task taking some random time to complete and giving some random reward.
Both random quantities are sampled from a priori unknown distributions.
In each round, the player may start a new task, as long as the set of active
tasks satisfies a constraint, which can be modeled as an arbitrary subset $A \subseteq 2^{[N]}$.

The problem is a generalization of the combinatorial semi-bandit
and the budget constrained bandit problems.
Assuming that each task takes only one time step gives the semi-bandit model while
assuming that at each time only a single task is active gives the budget constrained bandit problem where
the *cost* of each task is its time and the *budget* is the total time T.

The authors propose an algorithm regret $\tilde{O}(\sqrt{MNT})$ where $T$ is the time horizon,
$N$ is the number of tasks and $M$ is the maximum number of tasks that run at the same time.
Firstly, the authors show how to obtain a nearly optimal policy using the expected per-round rewards for tasks.
They then use a UCB-based approach on the per-round rewards to design the algorithm.

**Strengths:**

The paper is well-written studies an interesting problem which is a nice generalization of both combinatorial semi-bandits and the budget-constrained bandits.
The problem is well-motivated and the relation to related work is described very well.

**Weaknesses:**

The techniques in the paper seem to be mostly based on prior work,
though I think this is a minor point.

**Questions:**

In the beginning of section 3, the notation $b\le a$ is initially confusing as the tasks are thought
of as subsets rather than binary vectors. It would be better to add a short explanation to clarify this.
Formalize footnote in page 4.

Regarding the relation with blocking bandits,
I think it is also good to compare with the special case of task assignment when there are no constraints as this also very similar to blocking bandits.
In this case, seems to me that the key distinction between the problems is that in blocking bandits only a single action can be played in each round, while in bandit task assignments and arbitrary number
of tasks may be started at any moment.
Given that blocking bandits does not admit any pseudo-polynomial time algorithm, it may be good to mention why bandit task assignment can be *easier* than blocking bandits.

**Limitations:**

---

> ### Author Rebuttal · Authors · 2023-08-09
>
> > The techniques in the paper seem to be mostly based on prior work, though I think this is a minor point.
>
> Technical challenges in this paper are summarized in Lines 109--124. To cope with the challenge due to conflicting start and end times of tasks, we employed a phased-update approach, instead of simply choosing actions that maximize UCB as in CombUCB1 by [Kveton et al. (2015)]. To ensure that this approach works well, we need to carefully analyze the impact of the segment length $l_s$ on the regret and design $l_s$ appropriately, which is a main challenge in this work. Another innovation is to use Bernstein-type confidence bounds instead of standard Hoeffding-type confidence bounds, which is essential to achieve tight regret bounds as mentioned in Remark 5.2.
>
> > In the beginning of section 3, the notation $b\le q$ is initially confusing as the tasks are thought of as subsets rather than binary vectors. It would be better to add a short explanation to clarify this. Formalize footnote in page 4.
>
> Thank you for your suggestion.
> For vectors $b = (b_1, \ldots, b_N)^{\top}$ and $a = (a_1, \ldots, a_N)^{\top}$,
> $b \le a$ means that $b_i \le a_i$ holds for all $i \in \\{ 1, \ldots, N \\}$.
> We will add an explanation on this in the revised manuscript.
>
> > Regarding the relation with blocking bandits, I think it is also good to compare with the special case of task assignment when there are no constraints as this also very similar to blocking bandits. In this case, seems to me that the key distinction between the problems is that in blocking bandits only a single action can be played in each round, while in bandit task assignments and arbitrary number of tasks may be started at any moment. Given that blocking bandits does not admit any pseudo-polynomial time algorithm, it may be good to mention why bandit task assignment can be easier than blocking bandits.
>
> Thanks for your suggestion.
> In the task assignment problem with no constraints, the optimal strategy will be trivial, in which we make each task always processing, i.e., in each round, we start every task that is completed at that round.
> On the other hand, as the reviewer commented, blocking bandits only allow us to play a single action in each round, which makes the optimal strategy nontrivial.
> In fact, finding the optimal policy can be an NP-hard problem even if the true distributions of $r$ and $c$ are given.
> This is an illustrative example of why bandit task assignment can be easier than blocking bandits.
> The revised version will include such an explanation.

---

> > ### Comment · Reviewer_PHPg · 2023-08-15
> >
> > Thank you for you response; I don't have any additional questions. Given the clarifications I will raise my score.

---

### Official Review · Reviewer_uzGu · 2023-07-06

**Soundness:** 3 good
**Presentation:** 4 excellent
**Contribution:** 3 good
**Rating:** 6
**Confidence:** 2

**Summary:**

This paper introduces a novel bandit learning scenario, in which the learner can select in any time step tasks, each of which leads to stochastic rewards and processing time that blocks the task for several rounds. At any round, the tasks in progress (i.e., both newly started or still running ones) are constrained to be in a predefined given set of allowed tasks. The paper develops a solution to a regret minimization problem in this scenario and shows optimality of it by means of a lower bound.


**Strengths:**

The paper introduces an interesting and well-motivated learning scenario, which it distinguishes in detail from existing related ones. The developed solution to the corresponding regret minimization scenario comes with a non-trivial regret bound with a detailed proof in the appendix. Moreover, a corresponding lower bound indicates its optimality.
The paper is well-written in terms of used notation and the presentation of the proofs, it has very little typos.

**Weaknesses:**

- In Thm. 5.1, the gap-free bound is of the form $O(X + \frac{C_u^2}{C_l^2} NM\ln T)$. However, in the later discussion (e.g. in line 326), this second summand seems to be ignored. Why? Note that its dependence on $M, N$ and $C_u$ is worse than that of term $X$. And in line 355 you explicitly mention this term.
- As already mentioned by the authors, their solution requires knowledge of $C_u$ and $C_l$, which may not be possible in real-world. Can you discuss or test empirically the influence, which false beliefs of these values would have on the suffered regret?
- The empirical evaluation of the proposed solution is very limited. The authors do not consider real-world data, and the synthetic experiments are limited to two scenarios with the same $\bar{r}$. What about other values of $\bar{r}$? Would it make sense to evaluate the algorithms on randomly chosen $\bar{r}$, $\bar{c}$?

Typos:
- 216: $\min$ in the definition of $R_T$?
- 262: $1-8/t^2$?
- 352: wait

**Questions:**

See weaknesses.

**Limitations:**

The authors address the limitation of their work in the conclusion.

---

> ### Author Rebuttal · Authors · 2023-08-09
>
> > In Thm. 5.1, the gap-free bound is of the form $O(X + \frac{C_u^2}{C_l^2}NM \ln T)$. However, in the later discussion (e.g. in line 326), this second summand seems to be ignored. Why? Note that its dependence on $M$, $N$ and $C_u$ is worse than that of term $X$. And in line 355 you explicitly mention this term.
>
> In regret analysis (especially of lower bounds), asymptotic behavior for sufficiently large $T$ is often of primary interest. The second term ($\frac{C_u^2}{C_l^2}NM \ln T$) is smaller than $X = \tilde{O}(\frac{1}{C_l}\sqrt{C_u NMT})$ in cases where $T$ is sufficiently large, so it is omitted in several places for the sake of readability.
> Thanks to the reviewer's comment,
> however,
> we realized that the current description may cause confusion,
> so in the revised version,
> we will explicitly mention the $O(\ln T)$ terms in each case.
>
> > As already mentioned by the authors, their solution requires knowledge of $C_u$ and $C_l$, which may not be possible in real-world. Can you discuss or test empirically the influence, which false beliefs of these values would have on the suffered regret?
>
> Figure 5 in the newly uploaded file shows how false beliefs about $C_u$ affect the performance of the algorithms.
> Experimental results suggest that the performance of the proposed algorithm is sensitive to underestimation of $C_u$, while it is robust to overestimation of $C_u$.
> From this result, it is recommended that $C_u$ be set to as large a value as possible with a margin of error.
> The revised manuscript will include this observation.
>
> > The empirical evaluation of the proposed solution is very limited. The authors do not consider real-world data, and the synthetic experiments are limited to two scenarios with the same $\bar{r}$. What about other values of $\bar{r}$? Would it make sense to evaluate the algorithms on randomly chosen $\bar{r}$, $\bar{c}$?
>
> Thank you for this suggestion.
> Figures 1 and 2 in the newly uploaded file show the results of synthetic experiments for randomly chosen $\bar{r}$ and $\bar{c}$.
> This experiment confirms that the proposed algorithm works well empirically in randomly generated environments, including cases where the values of $\bar{r}$ are not necessarily constant.
> The revised manuscript will include the results of additional experiments.
>
> > Typos
>
> Thank you for taking the time to read our paper carefully.
> All of your points are correct and will be reflected in the revised version.

---

> > ### Comment · Reviewer_uzGu · 2023-08-15
> >
> > I thank you for your response, you have addressed all my concerns. Due to the clarifications and the further results I raise my score.

---

### Official Review · Reviewer_iLmP · 2023-07-06

**Soundness:** 3 good
**Presentation:** 3 good
**Contribution:** 3 good
**Rating:** 7
**Confidence:** 4

**Summary:**

This paper addresses thel problem of bandit task assignment, where a set of tasks need to be sequentially selected with uncertain processing times. The paper presents a UCB-type algorithm designed for bandit task assignment and provides theoretical analyses on upper bounds of the proposed algorithm and a regret lower bound of the problem. These bounds quantify the suboptimality of the algorithm and provide insights into its performance. The paper conducts numerical experiments to evaluate the proposed algorithm. It compares the proposed algorithm with existing approaches and demonstrates its superiority.

**Strengths:**

-	The paper investigates the bandit task assignment problem, which combines elements of task assignment and multi-armed bandit problems.
-	The paper provides a rigorous theoretical analysis of the proposed algorithm, deriving regret bounds and a gap-free regret bound. The analysis is well-founded and based on solid mathematical reasoning, ensuring the reliability and validity of the presented results.
-	The paper is well-written and well-organized. The concepts, algorithms, and theoretical analyses are presented in a clear and understandable manner. The paper also includes detailed explanations and supplementary material, enhancing the clarity and comprehensibility of the presented work.
-	By addressing the bandit task assignment problem, the paper may be useful in tackling a practical challenge that arises in various real-world scenarios. The proposed algorithm and its regret bounds offer a promising solution for optimizing task assignment decisions in the presence of uncertain processing times. The numerical experiments provide empirical evidence of the algorithm's superiority compared to existing approaches, further emphasizing its significance and potential impact in practice.


**Weaknesses:**

- There exist related works that have not been addressed in the work, including:
Learning to control renewal processes with bandit feedback, S Cayci, A Eryilmaz, R Srikant, 2019.
Budget-constrained bandits over general cost and reward distributions, S Cayci, A Eryilmaz, R Srikant, 2020.
-	The paper relies solely on synthetic datasets for the experimental evaluation, which may not fully capture the complexities and nuances of real-world scenarios. Incorporating real-world data or conducting experiments on real systems would provide a more comprehensive assessment of the algorithm's performance and validate its effectiveness in practical settings. Additionally, including case studies or application-specific examples would enhance the paper's relevance and demonstrate its potential impact in real-world contexts.
-	The paper lacks a detailed analysis of the algorithm's computational complexity. Understanding the time and space complexity of the proposed algorithm is crucial for assessing its scalability and efficiency. Providing an in-depth analysis or empirical evaluation of the algorithm's computational requirements would help readers understand its feasibility for large-scale problems and enable comparison with other existing approaches.
-	The paper does not provide a thorough sensitivity analysis of the algorithm's performance with respect to variations in key parameters. Understanding how the algorithm behaves under different settings, such as varying values of $C_u$, $C_1$, $N$, or $M$, would provide valuable insights into its robustness and generalizability. Conducting sensitivity analysis experiments and discussing the algorithm's performance across a range of parameter values would strengthen the paper's findings and help identify any limitations or constraints of the proposed approach.
-	The paper focuses primarily on the theoretical aspects of the bandit task assignment problem and the proposed algorithm. However, it provides limited guidance on the practical implementation and deployment of the algorithm in real-world systems. Discussing practical considerations, such as handling dynamic environments, adapting to changing task characteristics, or integrating with existing task assignment frameworks, would provide valuable insights and enable readers to apply the algorithm in practical scenarios.


**Questions:**

-	In the conclusion (Section 7), the authors briefly mentioned the potential modification of the problem framework to address the assumption of known upper and lower bounds on processing times. It would be helpful to elaborate on possible modifications or extensions that could relax this assumption. Specifically, how would allowing for uncertainty or adaptability in the processing time bounds impact the algorithm design and analysis? Discussing potential modifications or extensions would provide insights into the algorithm's adaptability to dynamic or uncertain environments.
-	The paper assumes that the bandit task assignment problem operates in a centralized setting where a single entity makes task assignment decisions. It would be helpful to consider or discuss potential extensions of the problem to a decentralized setting, where multiple agents collaborate to make task assignment decisions. Exploring the implications of decentralization and studying the design of distributed algorithms for task assignment could offer insights into real-world applications involving multiple decision-making entities.
-	(Minor comment) There is a missing period in line 267.


**Limitations:**

-	The authors address its potential limitations and future works in Section 7.

---

> ### Author Rebuttal · Authors · 2023-08-09
>
> > There exist related works that have not been addressed in the work, including: Learning to control renewal processes with bandit feedback, S Cayci, A Eryilmaz, R Srikant, 2019. Budget-constrained bandits over general cost and reward distributions, S Cayci, A Eryilmaz, R Srikant, 2020.
>
> Thank you for sharing related studies.
> Although these studies do not seem to address combinatorial constraints,
> we believe that they are certainly relevant as they deal with similar concepts as processing time.
> The revised version will definitely cite these papers.
>
> > The paper relies solely on synthetic datasets for the experimental evaluation, which may not fully capture the complexities and nuances of real-world scenarios. ...
>
> We agree that using real-world data can provide valuable insights into the complexities and nuances of practical scenarios.
> However, from the perspective of ensuring reproducibility, we consider that synthetic data would be more appropriate than real data.
> Since the proposed problem framework and algorithm are applicable to general problems that are not limited to specific applications, a common distribution (binomial distribution) with no special structure was employed in the experiments.
>
> > The paper lacks a detailed analysis of the algorithm's computational complexity. Understanding the time and space complexity of the proposed algorithm is crucial for assessing its scalability and efficiency. ...
>
> A brief note on the computational complexity can be found in Lines 281-285.
> The procedure of the proposed algorithm includes arithmetic operations and offline linear optimization over $\mathcal{A}$
> (As explained in Lines 203--209, we assume that we are given access to an offline linear optimization oracle).
> The number of arithmetic operations in each round is bounded by $O(N)$ since the update of each parameter in Step 7--10 of Algorithm 1 can be performed in $O(1)$ computation.
> Further, as discussed on Lines 281--285 of our paper,
> the number of calls to the offline linear optimization oracle is at most $O(\frac{C_u}{C_l} N \ln T )$,
> which is more efficient than standard UCB algorithms for combinatorial semi-bandits algorithms [23,24] that require $O(T)$ calls to the oracle.
> The space complexity, other than that required for the offline optimization oracle, is $O(N)$ since the algorithm works by maintaining $O(N)$ parameters.
> The code included in the supplementary is implemented to meet the above computational complexity.
> Figure 4 in the newly uploaded file shows the computation time for varying problem sizes,
> which suggests that the proposed algorithm is superior in terms of computational complexity.
> The revised version will include a more detailed explanation of the algorithm's computational complexity.
>
> > The paper does not provide a thorough sensitivity analysis of the algorithm's performance with respect to variations in key parameters. Understanding how the algorithm behaves under different settings, such as varying values of $C_u$, $C_l$, $N$, or $M$, would provide valuable insights into its robustness and generalizability....
>
> Some results of an empirical evaluation of the algorithm's sensitivity to parameters $N$ and $C_u$ are given in Figures 3, 4, and 5 in the newly uploaded file.
> Figure 3 suggests that the values of the regret scale linearly with respect to $N$, which is consistent with the regret upper bound of Eq.(7).
> Figure 5 represents how false beliefs about $C_u$ affect the performance of the algorithms.
> Experimental results suggest that the performance of the proposed algorithm is sensitive to underestimation of $C_u$, while it is robust to overestimation of $C_u$.
>
> > The paper focuses primarily on the theoretical aspects of the bandit task assignment problem and the proposed algorithm. However, it provides limited guidance on the practical implementation and deployment of the algorithm in real-world systems....
>
> While we agree that guidance such as that suggested by the reviewers would be beneficial to some readers,
> we have not included such guidance in this paper for the following reasons:
> As the reviewer is aware, the primary aim of this paper is to propose a novel theoretical framework rather than focusing on specific implementation methods. The potential techniques and protocols related to implementation can vary widely, and attempting to cover these details in this paper could lead to a diffusion of focus and a deviation from the main subject.
>
> > In the conclusion (Section 7), the authors briefly mentioned the potential modification of the problem framework to address the assumption of known upper and lower bounds on processing times. It would be helpful to elaborate on possible modifications or extensions that could relax this assumption. ...
>
> One possible modification would be to allow the player to suspend processing tasks.
> In this modified problem setup,
> even in cases where $C_u$ is incorrectly underestimated,
> it will be possible to immediately abort the task and correct $C_u$ when the underestimation is detected.
> We expect that this strategy and the appropriate parameter update rules for $C_u$ and $C_l$ allow us to remove the assumption of prior knowledge about processing times.
>
> > The paper assumes that the bandit task assignment problem operates in a centralized setting where a single entity makes task assignment decisions. It would be helpful to consider or discuss potential extensions of the problem to a decentralized setting, ...
>
> Thank you for this comment.
> The extension of the problem to a decentralized setting as you suggested is an interesting and useful direction for applications.
> This will be mentioned as future work in the revised version.
>
> > (Minor comment) There is a missing period in line 267.
>
> Thank you for pointing this out. We will modify it in the revised version.

---

> > ### Comment · Reviewer_iLmP · 2023-08-19
> >
> > I thank the authors for the thorough response. I believe my original rating captures my evaluation correctly, so it is unchanged.

---

### Official Review · Reviewer_2o4u · 2023-07-10

**Soundness:** 3 good
**Presentation:** 2 fair
**Contribution:** 4 excellent
**Rating:** 7
**Confidence:** 3

**Summary:**

The paper introduces a new bandit problem where at each round the agent choses a subset of tasks to run, where the length of tasks and their rewards are unknow and revealed only after the task is done. A UCB style algorithm is proposed and shown to achieve almost-optmial regret bounds.

**Strengths:**

1. The paper introduces a new bandit setting which generalizes quiet few other settings, so it has many potential applications in practice and theory.
2. The synthetic experiments demonstrate that the proposed algorithm immensely outperforms the related algorithms.
3. The analysis are well detailed and sketch proofs are included in the text.

**Weaknesses:**

1. The presentation could improve by moving the related works and side discussions to the later sections and instead focusing on the main setting introduction. It is hard to understand the setting since it is long laid in several pages.

**Questions:**

I suggest to summarize Section 1 and move Section 2 to after Section 5.


**Limitations:**

Yes.

---

> ### Author Rebuttal · Authors · 2023-08-09
>
> We sincerely appreciate the time and effort you have invested in this review.
>
> > The presentation could improve by moving the related works and side discussions to the later sections and instead focusing on the main setting introduction. It is hard to understand the setting since it is long laid in several pages. I suggest to summarize Section 1 and move Section 2 to after Section 5.
>
> Thank you for your suggestion.
> In the revised version, we will modify the structure of the text based on the feedback we received.

---

### Author Rebuttal · Authors · 2023-08-09

We deeply appreciate the reviewers' thoughtful and comprehensive comments and feedback.
The reviewers' insights have greatly contributed to the improvement of our work, and we sincerely appreciate the time and effort you have invested in this review.
We hope our response below addresses your concerns.

To respond to the comments by Reviewer uzGu and Reviewer iLmP, we performed additional experiment, of which results can be found in the newly uploaded PDF file.
In this PDF file, settings of parameters not specified in the captions are the same as in the reviewed manuscript (Section 5 and Appendix B).

---

### Decision · Program_Chairs · 2023-09-21

**Decision:**

Accept (poster)

**Comment:**

This paper studies the bandit task assignment problem, where a set of tasks needs to be sequentially selected with uncertain processing times. This paper proposes and analyzes a UCB-type algorithm, and conducts numerical experiments to evaluate the proposed algorithm.

All reviewers recommend accepting this paper. After reading the paper, the reviews, the rebuttals, and the discussions, I agree with the reviewers' decision.